

# Evaluation of denitrification from three biogeochemical models using laboratory measurements of N₂, N₂O and CO₂

Balázs Grosz, B.[1], Reinhard Well[1], Rene Dechow[1], Jan Reent Köster[1], M. Ibrahim Khalil[5], Simone Merl[1], Andreas Rode[4], Bianca Ziehmer[3], Amanda Matson[1], Hongxing He[2, 6]

[1]Thünen Institute of Climate-Smart Agriculture, Braunschweig, Germany
[2]University of Gothenburg, Department of Biological and Environmental Sciences, Gothenburg, Sweden
[3]Mühlhof, Jettenbach -Pfalz, Germany
[4]Ingenieurbüro Landwirtschaft und Umwelt (IGLU), Göttingen, Germany
[5]UCD School of Biology & Environmental Science, University College Dublin, and Prudence College Dublin, Ireland
[6]Swedish University of Agricultural Sciences, Department of Forest Ecology and Management, 901 83 Umeå, Sweden

*Correspondence to*: Balázs Grosz (balazs.grosz@thuenen.de)

**Abstract.** Biogeochemical models are useful for the prediction of nitrogen (N) cycling processes, but accurate description of the denitrification and decomposition sub-modules is critical. Current models were developed before suitable soil N₂ flux data were available; new measurement techniques have enabled the collection of improved N₂ data. We use measured data from two laboratory incubations to test the denitrification sub-modules of existing biogeochemical models. Two arable soils – a silt-loam and a sand – were incubated for 34 and 58 days, respectively. Fluxes of $N_2$, $N_2O$ and $CO_2$ were quantified using gas chromatography and isotope-ratio mass spectrometry (IRMS). For the loamy soil, seven moisture and three $NO_3^-$ contents were included, with temperature changing during the incubation. The sandy soil was incubated with and without incorporation of litter (ryegrass), with temperature, water content and $NO_3^-$ content changing during the incubation. Three common biogeochemical models (Coup, DNDC and DeNi) were tested using the data. No systematic calibration of the model parameters was conducted since our intention was to evaluate the general model structure or 'default' model runs. As compared with measured fluxes, the average $N_2+N_2O$ fluxes of the default runs for loamy soil were approximately 3 times higher for Deni, 105 times smaller for DNDC and 22 times smaller for Coup. For the sandy soils, default runs were 3 times higher for DeNi, 7 times smaller for DNDC and 12 times smaller for Coup. While measured fluxes were overestimated by DeNi and underestimated by DNDC and Coup, the temporal patterns of the measured and the modeled emissions were similar for the different treatments. None of the models was able to determine litter-induced decomposition correctly. The reason for the differences between the measured and modeled values can be traced back to model structure uncertainty and/or parameter uncertainty. Given the aim of our work - to assess existing model processes for further development and/or to identify missing processes within the models - these results provide valuable insights into avenues for future research. We conclude that the predicting power of the models could be improved through future experiments that collect data on denitrification activity with a concurrent focus on control parameter determination.



# 1 Introduction

Although our understanding of nitrous oxide (N$_2$O) fluxes, nitrogen (N) use efficiency and N leaching in agricultural
ecosystems has steadily increased in recent decades (Galloway et al., 2004; Singh 2011; Zaehle 2013), we still have only a
limited understanding of soil denitrification and the complex interaction of factors controlling denitrification processes.
Addressing this knowledge gap is crucial for mitigating nitrogen fertilizer loss as well as for predicting and reducing N$_2$O
emissions.

Denitrification is an anaerobic soil process by which microbes carry out the step-by-step reduction of nitrate (NO$_3^-$), to nitric
oxide (NO), N$_2$O and finally dinitrogen (N$_2$) (Groffman et al., 2006). The production and consumption of N$_2$O via
denitrification is affected by: temperature (Rodrigo et al., 1997), O$_2$ concentration (Müller and Clough 2014), moisture
(Grundmann and Rolston 1987; Groffman and Tiedje 1998), pH (Peterjohn 1991; Simek and Hopkins 1999; Simek and
Cooper 2002), substrate (N oxides and organic carbon ) availability (Heinen 2006) and gas diffusivity (a function of water
content) of the soil (Leffelaar 1988; Leffelaar and Wessel, 1988; Li et al., 1992a; Del Grosso et al. 2000; Schurgers et al.,
2006). Field measurements of denitrification that explore the interactions between these factors are challenging, due to the
methodological issues surrounding the measurement of N$_2$ fluxes (high background N$_2$ and low soil N$_2$ flux) (Groffman et
al., 2006). However, the impact of these different factors on denitrification can be assessed with properly designed
laboratory experiments (Butterbach-Bahl et al., 2013; Cardenas et al., 2003).


Models are an important tool to explore complex interactions and develop climate-smart strategies for agriculture
(Butterbach-Bahl et al., 2013). Although numerous models exist, which predict denitrification in varying environments and
at different scales (Heinen 2006), it has always been challenging to evaluate the accuracy of modeled denitrification due to
the paucity of suitable measured data (Sgouridis et al., 2016, Scheer et al., 2020). Simplified process descriptions, inaccurate
model parameters and/or inadequately collected input data result in poor predictions of N$_2$ and N$_2$O fluxes (Parton et al.,
1996). While in many studies N$_2$O emissions alone are used to develop and train models (Chen et al., 2008), measurements
of both N$_2$O and N$_2$ fluxes, under varying soil conditions, are necessary to develop and test model algorithms (Li et al., 1992;
Parton et al., 1996; Del Grosso et al., 2000). Data suitable to validate N$_2$O and N$_2$ flux calculations within denitrification sub-
modules are still scarce, yet these large datasets are needed in order to validate models and improve their accuracy with
respect to denitrification processes.

Three robust, well-used models for describing denitrification processes are: Coup (Jansson and Moon, 2001), DNDC (Li et
al., 1992) and DeNi (based on the approach of the NGAS and DailyDayCent; Parton et al., 1996 and Del Grosso et al.,
2000). These models were developed between 20 and 30 years ago and, with minor modifications, are still used today.
Within each of the three models (Coup, DNDC, DeNi), the denitrification sub-modules use different approaches to address

the complexity of denitrification, including how they consider controlling factors (e.g. soil moisture, heat transfer, nitrification, decomposition, growth/death of the denitrifiers) as well as how they simulate temporal and spatial dynamics. Despite the success with which each of these models has been used, the incorporation of recent advancements in our understanding of denitrification may be able to improve model estimates. For example, the development of the NGAS and

DailyDayCent models used measured denitrification data based on the acetylene inhibition technique (Weier et al., 1993; Parton et al., 1996 and Del Grosso et al., 2000), which is no longer considered suitable for many applications (Bollmann and Conrad, 1997; Nadeem et al., 2013; Sgouridis et al., 2016). The lack of the proper $N_2$ datasets, and new research not being integrated into existing models, has developed into an urgent need for focused model development using newly developed and/or more precise data collection techniques.


In this study, we use newly measured data to test the denitrification products simulated by existing biogeochemical models as a pre-requisite for the development of new or improved approaches to denitrification modelling. Our aim in this study was not to fit the magnitude of the modeled fluxes to the measured values and rate the performance of the models. Instead, we aim to identify missing processes and limitations in the denitrification sub-modules and determine the best next step for

model development. Therefore, the denitrification sub-modules of the models were not calibrated, since the calibration would have been different for the different experimental settings and the results of the model runs would not have been comparable for the different measurements. Without calibration, we can compare the performance of the sub-modules with the same (factory) settings for the different experimental treatments. Specifically, our aims were to: (i) compile and present unpublished $N_2$ and $N_2O$ results from two laboratory incubations (Ziehmer, 2006, Merl, 2018) (ii) simulate denitrification

products using the three models (Coup, DNDC, DeNi) (iii) compare the measured and modeled values, (iv) identify soil conditions when models could or could not predict $N_2$ and $N_2O$ fluxes, and (v) make suggestions for model improvement.

## 2 Materials and methods

### 2.1 Denitrification data collection

#### 2.1.1 Hattorf field site (silt-loam soil)

Soil samples were taken in October 2005 from an arable soil near Hattorf (hereafter referred to as the silt-loam soil), Lower Saxony, Germany, in the loess-covered Pöhlde basin near the Harz mountains (51°39.35868' N, 10°14.71872' E, 215 m a.s.l.). The site is in the transition zone of the cool continental/subarctic climate and warm-summer humid continental climate, where the mean annual temperature is between 7 and 8.5°C and the average yearly precipitation is 700 mm. The cropping rotation of the site was winter rape – winter wheat – winter barley, and sampling was conducted when the

vegetation was winter rape. The Haplic Luvisol soil had a silt-loam texture with relatively low organic matter content (Table 1). In the field, a 4 m² area was marked out for sampling. In this area, plants (winter rape) were first removed and then





surface soil (0 to 10 cm depth) was collected in large, plastic boxes. Soil was returned to the lab, where it was sieved to 10 mm, homogenized, subsamples analyzed for physical and chemical properties (Table 1), and remaining soil stored at 4ºC until use.


Table 1: Physical and chemical data of surface soil from Hattorf (silt-loam, 0 to 10 cm depth) and Fuhrberg (sand, 5 to 20 cm depth), Germany

| | Clay | Silt | Sand | Bulk density | $NO_3^-$ | $NH_4^+$ | pH | Total N | Organic C | C/N ratio |
|---|---|---|---|---|---|---|---|---|---|---|
| | [%] | [%] | [%] | [g/cm$^3$] | [mg N kg$^{-1}$] | [mg N kg$^{-1}$] | (CaCl2) | [%] | [%] | |
| Hattorf | 15.2 | 77.6 | 7.2 | 1.4 | 12.73 | 1.27 | 6 | 0.1 | 1.05 | 10 |
| Fuhrberg | 3.1 | 5.9 | 91.0 | 1.5 | 17.8 | 1.4 | 4.8 | 0.1 | 2.1 | 15.5 |

### 2.1.2 Fuhrberg field site (sand soil)

Soil samples were taken in August 2016 from an arable soil near Fuhrberg (hereafter referred to as the sand soil), Lower
Saxony, Germany (52°33.17622' N, 9°50.85816' E, 40 m asl). The site is in the transition zone of the temperate oceanic climate and warm-summer humid continental climate, where the mean annual temperature is 8.2°C and the average yearly precipitation is 680 mm. Typical crops during the preceding decades were winter cereals, potatoes, sugar beet and maize. The soil is a Gleyic Podzol developed in glacifluvial sand (Böttcher et al., 1999; Well et al., 2005). The first 5 cm of soil contained incorporated winter wheat straw residuals. To avoid inaccuracy in the measurement of soil parameters, this 5 cm
layer was removed by hand in a 100 m$^2$ area followed by the collection of soil from a depth of 5 to 20 cm. After field collection, soil was transported to the lab, air dried, sieved to 10 mm, homogenized and stored in plastic boxes at 4°C until use. The measured physical and chemical properties of the soil are shown in Table 1.

### 2.1.3 Silt-loam laboratory incubation

To avoid measuring the effect of rewetting (increased respiration and mineralization) during the incubation, soil was pre-
incubated at room temperature for 2 weeks at 50% of maximum water holding capacity. After the pre-incubation period, $^{15}$N-KNO$_3$ solutions (see Tables 2 and 3 for concentrations) were added to subsets of soil and thoroughly mixed. Soils were then packed into plexiglass cylinders (14.4 cm diameter) at typical field bulk density (1.4-1.5 g cm$^{-3}$) and a soil depth of 25 cm. Distilled water was then added to each cylinder to bring the water-filled pore space (WFPS) up to the target for each treatment (Table 2). The soil cylinders were incubated for 34 days, during which the headspace was continuously flushed
with ambient air at a flow rate of 6 ml min$^{-1}$. During the incubation, only temperature was changed (Fig. S.3), while the initial settings of water content were kept constant. Temperatures were selected to mimic winter conditions, to assess





whether previously observed $NO_3^-$-N losses during winter could be explained by denitrification (Ziemer, 2006). Gas samples were collected manually once a day and analyzed by gas chromatography (GC) (Well et al., 2009) to determine $N_2O$ and $CO_2$ fluxes, and by isotope ratio mass spectrometry (IRMS) to determine the flux of $N_2+N_2O$ originating from the $^{15}N$–

labeled $NO_3^-$ (Well et al., 1998; Lewicka-Szczebak et al., 2013). Soil samples were collected at the beginning and end of the incubations and analyzed for $NO_3^-$, $NH_4^+$ and water content as described in Buchen et al. (2016).

Table 2: Initial settings of laboratory incubations of soil from Fuhrberg (Sand) and Hattorf (silt-loam; treatments I to VII), Germany.

| | Silt-loam | | | | | | | Sand |
|---|---|---|---|---|---|---|---|---|
| | I | II | III | IV | V | VI | VII | |
| Added N ($KNO_3$) [mg N kg$^{-1}$ dry soil] | 20 | 10 | 40 | 20 | 20 | 20 | 20 | 50 |
| atom % $^{15}N$ in $KNO_3$ | 60 | 98 | 60 | 60 | 60 | 60 | 60 | 60 |
| Thickness of soil layer (cm) | 25 | 25 | 25 | 25 | 25 | 25 | 25 | 10 |
| Bulk density | 1.4 | 1.4 | 1.4 | 1.46 | 1.52 | 1.4 | 1.4 | 1.5 |
| grav. water content | 0.25 | 0.27 | 0.27 | 0.25 | 0.25 | 0.27 | 0.30 | 0.231* |
| WFPS (%) | 73.0 | 80.0 | 80.0 | 80 | 88 | 80.0 | 90.0 | 80.0* |


Table 3: Initial extractable N content, N fertilization and calculated $^{15}N$ enrichment after fertilization in laboratory incubations from Fuhrgerg (sand) and Hattorf (silt-loam), Germany. The Hattorf incubation had 7 different treatments (Roman numbers) with 3 different added N values.

| Experiment (variants) | $NO_3^-$-N + $NH_4^+$-N in the unfertilized soil [mg N kg$^{-1}$ dry soil] | Added N ($KNO_3$) [mg N kg$^{-1}$ dry soil] | Calculated $^{15}N$ enrichment (at%) of the $NO_3^-$ in the soil |
|---|---|---|---|
| Silt-loam (II) | 14 | 10 | 41 |
| Silt-loam (I, IV, V, VI, VII) | 14 | 20 | 35 |
| Silt-loam (III) | 14 | 40 | 45 |
| Sand | 16 | 50 | 60 |



### 2.1.4 Sand laboratory incubation

To exclude the phase of intensive respiration and mineralization typically following rewetting, soils were pre-incubated at 50% of maximum water holding capacity for 3 weeks (at room temperature). After pre-incubation, $^{15}N$-labelled $KNO_3$ solution (50 mg N $kg^{-1}$ dry soil) was added to the soil, and thoroughly mixed (Table 2 and Table 3). After addition of $NO_3$, the soil was divided, and in half of it, ground ryegrass (sieved with 1 mm mesh; added at a rate of 2.2 g $kg^{-1}$ dry soil) was also homogenously incorporated. The ryegrass had a C-to-N ratio of 25, and nitrogen, carbon and sulphur content of: 1.3%, 32.2% and 0.4%, respectively. Four replicates of soil from each of the two treatments (with and without ryegrass) were then packed into plexiglass cylinders (14.4 cm inner diameter) at typical field bulk density (1.5 g $cm^{-3}$) and a soil depth of 10 cm (Table 2). The soil cylinders were incubated for 58 days, during which the headspace was continuously flushed with an artificial gas mixture (2% $N_2$ and 20% $O_2$ in He) at a flow rate of 20 ml $min^{-1}$. The low $N_2$ concentration was established to increase the sensitivity of $N_2$ flux detection (Lewicka-Szczebak et al., 2017).

The cylinders were incubated using an automated incubation system, including gas analysis by GC, suction plates at the bottom of the cylinders to control water potential and collect leachate, and an irrigation device to mimic precipitation and/or fertilization. Using the GC, $N_2O$, $CH_4$, $N_2$ and $CO_2$ (Säurich et al., 2019) were continuously measured throughout the incubation. Gas samples were also collected manually for IRMS analysis, to determine fluxes of $N_2$ and $N_2O$ originating from the $^{15}N$ labeled $NO_3^-$ (Well et al., 1998; Lewicka-Szczebak et al., 2013). The pressure head at the suction plates was controlled by connecting the bottles for seepage collection to a gas reservoir, which was maintained at the target pressure ($\pm0.5$ kPa). Water potential in the soil column resulted from the difference in the pressure head between the soil cylinder headspace and suction plate. Headspace pressure was positive due to the continuous headspace flow and flow restriction in the exhaust line of the gas sampling system. Instability in the headspace pressure (values between 1 and 3 kPa) occurred near the end of the experiment, due to partial clogging of the hypodermic needles that were used to lead the exhaust gas through sampling vials (Well et al., 2006). Therefore, pressure head in the soil columns was associated with an uncertainty of about 2.5 kPa.

The water content of the soil was initially set to 0.231 g $g^{-1}$ (equivalent to 80 % WFPS) and was subsequently changed by establishing defined water potential at the suction plates (Table 4) and by adding water and/or $KNO_3$ solution from the top of the columns as irrigation/fertilization events. Phases with defined temperature were set as shown (Fig. S.3 and Table 4). Soil samples were collected at the beginning and end of the incubations and analyzed for $NO_3^-$, $NO_2^-$, $NH_4^+$, DOC, pH and water content as described in Buchen et al. (2016).

Table 4: Experimental settings during an 8-week laboratory incubation of re-packed soil cores from Fuhrberg, Germany.

| Week of Experiment | 1 | 2 | 3 | 4 | 5 | 6 | 7 | 8 |
|---|---|---|---|---|---|---|---|---|




| | | | | | | | | |
|---|---|---|---|---|---|---|---|---|
| Bottom water potential (kPa) | -10 | -20 | -60 | -60 | -10 | -10 | -10 | -10 |
| Temp. °C | 20 | 20 | 20 | 20 | 20 | 10 | 5 | 10 |
| Irrigation with water (mm) | - | - | - | - | 10 | - | - | - |
| Irrigation with $NO_3^-$ solution [mm / mg N kg$^{-1}$] | - | - | - | - | 30 / 30 | - | - | - |

165

## 2.2 Model choice and description

Using the denitrification data collected in the incubations described above, we tested the denitrification sub-modules of three biogeochemical models: Coup (Jansson and Moon, 2001), DNDC (Li et al., 1992) and DeNi (based on the approach of the NGAS and DailyDayCent; Parton et al., 1996 and Del Grosso et al., 2000). Our first criterion of denitrification sub-modules evaluation was the agreement of measured and modeled results with respect to directional changes of $N_2$ and $N_2O$ (i.e. fluxes increasing or decreasing) in response to the relevant control factors. Comparing the magnitude of measured and modeled fluxes was not considered as a criterion. We note that individual control factors were tested only to a limited extent, and were otherwise affected by interactions with other control factors (see description of experimental design in 2.12.1). Nevertheless, comparing the temporal dynamics of fluxes measured in the experiments and given by the models, reveals the suitability and deficiencies of both, which can pave the way for planning better model evaluation experiments and for developing model routines to fill the gaps of the current approaches.

### 2.2.1 Coup

Coup (coupled heat and mass transfer model for soil–plant–atmosphere systems) is a complex, adjustable process-oriented model that uses a modified approach of PnET-N-DNDC to simulate nitrification and denitrification (Norman et al., 2008). It is a developed version of the SOIL and SOILN models (Jansson and Moon, 2001). The main model structure is a vertical layered, 1-D soil profile. Coup includes all main heat and water flow processes in the soil profile as well as exchange with the atmosphere.

A first order kinetics approach for two pools (litter and humus) governed by response functions of soil moisture and temperature is used to simulate soil organic carbon dynamics. In Coup, soil litter represents the rapidly decomposable organic material (e.g. fresh plant litter) and the humus pool represents the more resistant fraction. Fluxes of NO, $N_2O$ and $N_2$ are modeled via nitrification and denitrification, which in turn are obtained from modeled parameters including respiration, mineral N, and dissolved organic C (DOC). The soil anaerobic fraction is defined by the approach of the anaerobic balloon concept of DNDC (Norman et al., 2008).



The simulation of nitrification is calculated by the response functions of soil temperature and moisture, pH and $NH_3$

concentration (Norman et al., 2008) Denitrification processes are simulated by soil temperature, pH and the N concentration of the microbial pool and the anaerobic fraction of the soil (Jansson and Karlberg, 2011) (see Table S.6). The model can simulate C, N and water fluxes in hourly resolution. The complex modular structure gives flexibility to users for planning a step-by-step increase in the complexity of simulations. This option is ideal for the simulation of laboratory experiments. Users can freely define the thickness and the number of soil layers and the setup of the initial conditions of each layer. The

model can also simulate changes of the parameters between soil layers.

### 2.2.2 DeNi

DeNi was programmed based on the nitrification and denitrification approach of the NGAS model (an early stage of the DayCent model) (Parton et al., 1996) (see Table S.6). The approach of the DailyDayCent (and therefore DeNi) model for the description of denitrification is a hybrid between detailed process-oriented models and simpler nutrient cycling models

(Parton et al., 1996). It allows users to separately test the nitrification and denitrification sub-modules. The model runs on daily time steps. The main difference between DailyDayCent and Coup is that Coup, like other more complex process-oriented models, explicitly models denitrifier dynamics. In contrast, the DailyDaycent/NGAS model is a relatively simple, semi-empirical model to simulate the $N_2+N_2O$ production without directly considering microbes. It uses empirical parameters and functions that have no direct physical, chemical or biological explanation and were developed from

experimental observations. Therefore, it is the combination of a simplistic nutrient cycle model and a more detailed process-based model (Parton et al., 1996).

The $N_2O$ flux from nitrification is modeled using: soil pH, soil temperature, soil moisture, soil $NH_4^+$ concentration (available $NH_4^+$ is then computed as a function of $NH_4^+$ concentration), and the N turnover coefficient, which is a soil-specific parameter.

The denitrification sub-module calculates the fluxes of $N_2$ and $N_2O$. The soil heterotrophic respiration rate (depending on the available carbon), soil $NO_3^-$ concentration and soil moisture (WFPS) control total denitrification. The $N_2/N_2O$ ratio is calculated as a function $(F(NO_3/CO_2))$ of electron donor $(NO_3^-)$ to substrate and soil water content (Del Grosso et al., 2000).

### 2.2.3 DNDC

The Denitrification-Decomposition model (DNDC) is a complex, widely used process-based model of C and N

biogeochemistry in agricultural ecosystems (e.g. Li et al. 1994). It has been extensively tested globally and has shown reasonable agreements between measured and modeled $N_2O$ emissions for many different ecosystems (e.g. Li, 2007; Kurbatova et al., 2009; Giltrap et al., 2010; Khalil et al., 2016; 2018; 2019). Several modifications/versions have been developed to fit different ecosystems and those provide variable estimates depending on the model versions used. DNDC contains six sub-modules: soil climate, crop growth, decomposition, denitrification (see Table S.6), nitrification and

fermentation. It additionally includes subroutines for cropping practices (fertilization, irrigation, tillage, crop rotation and



manure addition). The model joins denitrification and decomposition processes together to predict emissions of C and N from agricultural soils, based on various soil, climate and environmental factors. It considers the soil as a series of discrete horizontal layers with uniform soil properties within each layer, except for some soil physical properties that are anticipated as being constant across all layers. However, time-dependent variations in soil moisture, temperature, pH, C and N pools are

considered for a reliable estimate of C and N fluxes by calculating them for each soil layer for each time step. Like in Coup, denitrifiers are explicitly modeled.

## 2.3 Model initialization

Selected experimental data for model evaluation included denitrification ($N_2$ and $N_2O$ fluxes produced from soil $NO_3^-$) and

"proximal" and "distal" controls (according to the definition by Groffman et al., 1988). Proximal controls were temperature, $NO_3^-$, pH and organic C. Distal controls were soil moisture, texture, $NH_4^+$-N, bulk density and respiration (as a proxy for $O_2$ consumption).

Our first criterion of model evaluation was the agreement of measured and modeled results with respect to directional

changes of $N_2$ and $N_2O$ (i.e. fluxes increasing or decreasing) in response to the relevant control factors. Comparing the magnitude of measured and modeled fluxes was considered a secondary criterion. In the two experimental setups, individual control factors were only tested to a limited extent, while the remaining measurements reflected the interaction of multiple control factors (see 2.1.3 and 2.1.4). Those interactions presented additional complexity, which would not classically be used for model evaluation, yet provided valuable data on the temporal dynamics of measured vs modeled fluxes (see discussion

for details).

Models were set up according to the initial experimental setups of the two incubations (i.e. 7 initial model set-ups for silt-loam and 2 set-ups for sand; Table 2). For the silt-loam soil, only soil temperature was changed during the experiment, while for the sand soil temperature, soil water status (change of the water potential and irrigation) and $NO_3^-$ content (by irrigation

with $KNO_3$ solution) were changed. We first compare to which extent the models fit the magnitude of fluxes in general, and subsequently, whether the models reflect the observed differences between the experimental treatments.

### 2.3.1 Coup

Coup gives users the option to choose between different algorithms, each representing the functionality of a sub-module,

with each sub-module addressing a different aspect of the soil-atmosphere-vegetation system (Senapati et al., 2016; He et al., 2016; Norman et al., 2008; Nylinder et al., 2011; Conrad and Fohrer, 2009). This feature was used to adapt the model structure to the experimental setup and the available data (Table S.7).



In the model, soil columns of sand were divided into 5 layers (we are assuming equilibrium, and it was calculated based on the water retention curve and layer depth) with layer extents of 2 cm. The water retention curve was not available for the silt-loam soil. The soil columns were thus modeled as a 25 cm unified, single soil layer. Daily water content and soil temperature were set up in the model as dynamic input parameters coming from water balances and measurements, respectively. The initial contents of organic carbon, total N, $NO_3^-$-N and $NH_4^+$-N of the silt-loam and sand were set up in the model (Table S.8). The initial amount of SOC allocated into the labile pool was based on default SOC allocation fractions. For sand treatments with application of ryegrass, the C and N of ryegrass were exclusively added to the labile pool. Since the basic settings resulted in overestimation of $CO_2$ production, first order decomposition rate coefficients for litter and humus were changed to modify decomposition and mineralization to fit measured respiration rates.

From the two available algorithms to describe denitrification, the algorithm with explicit consideration of denitrifiers was chosen (Table S.7), because we wanted to test a model which includes the microbial approach for the denitrification sub-model. The structure and the complexity of Coup made it necessary to modify some model parameters and settings to improve the fit between modeled and measured $N_2O$ and $N_2$ fluxes. The applied settings and parameters are in Tables S.6, S.7 and S.8. Parameters were adjusted separately for each experiment (silt-loam and sand) but were identical between treatments.

### 2.3.2 DeNi

Parameter adjustment and data input were accomplished using the DeNi source code. Measured soil texture, bulk density, initial $NO_3^-$, $NH_4^+$ and SOC were used to initialize the model. We ran the model calculated with one soil layer for the silt-loam soil and with five, 2 cm thick soil layers for the sand soil, with differing water contents. We used the measured daily temperature and the theoretical (calculated) water content of each of the 5 layers (see 2.3.1). Irrigation, seepage and fertilization events were included, and the model was modified with calculated changes in $NO_3^-$-N and water content, which were calculated based on the irrigation, seepage and fertilization events. The theoretical $NH_4^+$ and $NO_3^-$ concentrations (Table S.2) were changed (modeled production and consumption) by mineralization, nitrification, denitrification, leaching and the added fertilizer (Table S.1) during the simulations. For the calculation of missing soil physical parameters (e.g the soil gas diffusion coefficients) the respective pedotransfer functions were applied (Saxton and Rawls, 2000). Besides $N_2O$ and $N_2$, the model also calculated the soil fluxes of $CO_2$.

### 2.3.3 DNDC

The latest version of DNDC (DNDC95) was used to simulate $N_2O$, $N_2$ and soil $CO_2$ emissions. The model was originally designed for field and regional scales. Therefore, certain adjustments had to be made to establish suitable model inputs to



represent the conditions of the laboratory incubations. Based on the experimental setup for the sand soil, the irrigation with

$KNO_3$ solution had to be simulated as rainfall containing $NO_3^-$ and the atmospheric background of $NH_3$ and $CO_2$ was considered zero and negligible, respectively, since the incubation was in an artificial atmosphere. Minimum and maximum temperatures were set according to the actual experimental values. Measured soil inputs were included with microbial activity index (factor to modify the denitrification process) of 1. The mixing of the experimental soil prior to incubation was applied as litter-burying till with no crop and coupled with water and $NO_3^-$ fertilizer addition. Nitrate fertilizer was added

twice with ryegrass residue as straw either mixed or omitted. Water was added once in the beginning and twice in the middle of the experiment as per treatments in the form of irrigation following comparative tests with rainfall as well as rainfall and irrigation options.

To run the model using inputs from the silt-loam incubation, the microbial activity index, temperature setting and mixing of soil with water as irrigation and fertilizer were simulated as in the sand incubation but irrigation and fertilization were

assumed to occur only once in the beginning and rainfall was considered zero.

## 2.4 Statistics

Statistical calculations were done using the Python 3 (Van Rossum and Drake, 2009) and the R (R Core Team, 2013) programming languages and GNUPlot (Williams and Kelley, 2011) interactive plotting program. A multiple comparison of

means (Tukey HSD, $p<0.05$) was performed on the $N_2+N_2O$ and $CO_2$ data of the silt-loam soil. The $N_2+N_2O$ data of the sand soil was not normally distributed. Therefore, the Wilcoxon signed-rank test was used for these data to test the effect of the ryegrass application.

## 3 Results

### 3.1 Silt-loam soil

Results of the seven treatments are shown in Table 5. $CO_2$ fluxes were positively correlated with temperature (Fig. S.3). Cumulative $CO_2$ fluxes were generally highest in the treatments with low WFPS and lowest in the treatments with high WFPS and bulk density (Table 5; Fig. S.4). $N_2+N_2O$ fluxes decreased over time in treatments $I_{20N\_73\%\_1.4}$, $II_{10N\_80\%\_1.4}$, $IV_{20N\_80\%\_1.46}$, $VI_{20N\_80\%\_1.4}$ whereas the opposite was the case in treatments $III_{40N\_80\%\_1.4}$, $V_{20N\_88\%\_1.52}$, and $VII_{20N\_90\%\_1.4}$ (Fig.

S.1). Cumulative $N_2+N_2O$ fluxes decreased in the order $V_{20N\_88\%\_1.52}$ > $III_{40N\_80\%\_1.4}$ > $IV_{20N\_80\%\_1.46}$ > $I_{20N\_73\%\_1.4}$ > $VII_{20N\_90\%\_1.4}$ > $VI_{20N\_80\%\_1.4}$ > $II_{10N\_80\%\_1.4}$. The highest cumulative $N_2+N_2O$ fluxes were thus related to higher bulk density and WFPS (Table 5). The treatment with lowest $NO_3^-$ application ($II_{10N\_80\%\_1.4}$) showed the lowest $N_2+N_2O$ flux, while the highest bulk density resulted in higher $N_2+N_2O$ flux compared to all other treatments (Table 5). The $N_2O/(N_2+N_2O)$ ratio was generally low (between 0.088 and 0.264, Table 5).






Table 5: Measured cumulative fluxes ($N_2$, $N_2O$, $N_2+N_2O$: g N m$^{-2}$ day$^{-1}$; $CO_2$: g C m$^{-2}$ day$^{-1}$) and $N_2O/(N_2+N_2O)$ ratio of cumulated fluxes (dimensionless) - over 34 days – of 7 different treatments, during a laboratory incubation of arable, silt-loam soil from Hattorf, Germany. Shown in the treatment headings are added N (10/20/40 mg $KNO_3$-N / kg dry soil), water-filled pore space (73-90%) and bulk density. (1.4-1.52 g cm$^{-3}$).

|  | I | II | III | IV | V | VI | VII |
|---|---|---|---|---|---|---|---|
|  | 20N_73%_1.4 | 10N_80%_1.4 | 40N_80%_1.4 | 20N_80%_1.46 | 20 N _88%_1.52 | 20N_80%_1.4 | 20N_90%_1.4 |
| $N_2$ | 0.118 | 0.042 | 0.156 | 0.114 | 0.278 | 0.049 | 0.064 |
| $N_2O$ | 0.019 | 0.004 | 0.056 | 0.026 | 0.055 | 0.009 | 0.017 |
| $N_2+N_2O$ | 0.137[c] | 0.046[c] | 0.212[ab] | 0.140[bc] | 0.334[a] | 0.058[c] | 0.081[bc] |
| $N_2O/(N_2+N_2O)$ | 0.139 | 0.088 | 0.264 | 0.184 | 0.166 | 0.148 | 0.207 |
| $CO_2$ | 1.622[a] | 1.142[a] | 0.368[bc] | 1.041[ab] | 0.158[c] | 1.483[a] | 0.190[c] |


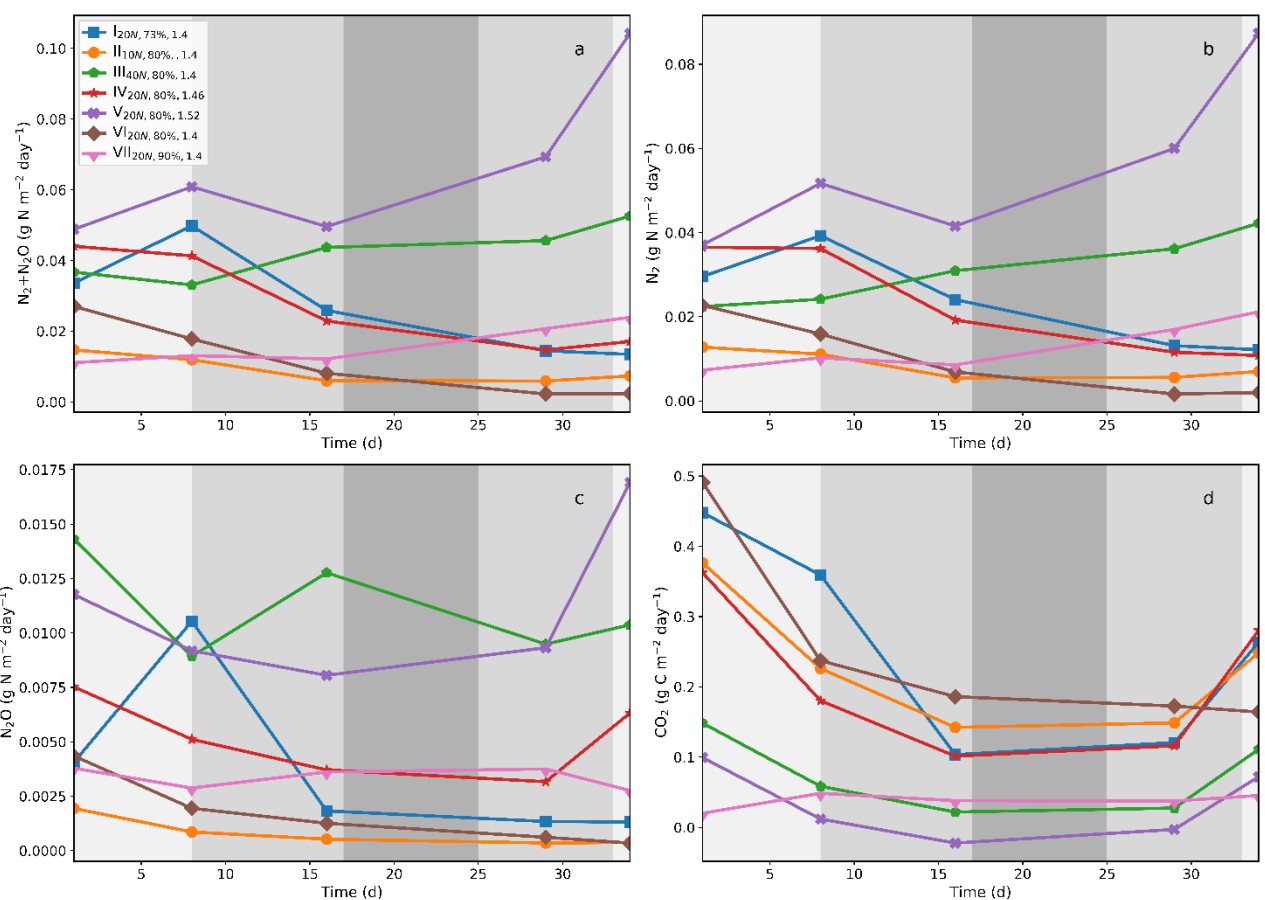

**Figure 1 a-d: Measured fluxes of N₂+N₂O (a), N₂ (b), N₂O (c) and CO₂ (d) of an arable, silt-loam soil from Hattorf, Germany (values shown are the mean of four replicates over a 34 days laboratory incubation). The background colors show the temperature during each time period (light grey: 10°C, middle grey: 6°C, dark grey: 2°C).**


### 3.2 Sand soil

Fluxes are shown for individual replicates of both treatments (Fig. 2 and Fig. 3), as variable pressure (see section 2.1.4) resulted in differing water content within and between treatments (Table S.1; Fig. S.4). The initial water content of 80% WFPS was equivalent to a water potential of -3 kPa according to the water retention curve of this soil (Fig. S.2). Leaching

dynamics were also highly variable between replicates (Table S.1).

Table 6: Measured cumulative fluxes (N₂, N₂O, N₂+N₂O: g N m⁻² day⁻¹; CO₂: g C m⁻² day⁻¹) and product ratios over a 58 days laboratory incubation of sandy soil from Fuhrberg, Germany. Shown are averages and standard deviation of 4 replicate cores with (C1-4) and without (C5-8) added ryegrass.




| | $N_2$ | $N_2O$ | $N_2+N_2O$ | $N_2O/(N_2+N_2O)$ | $CO_2$ |
|---|---|---|---|---|---|
| C1-4 | 0.490±0.075 | 4.82±0.632 | 5.31±0.677 | 0.908 | 54.6±0.646 |
| C5-8 | 0.053±0.005 | 0.638±0.097 | 0.691±0.100 | 0.924 | 15.1±0.136 |

Comparing the cumulative $CO_2$ fluxes of the two treatments, ryegrass-amended columns were (2-4 times) higher than those without ryegrass (Table 6). The $CO_2$ fluxes reached a maximum after 8-13 days and then slightly decreased until the Day 32 (Fig. 2d), when both irrigation (Fig. S.3) and temperature (Fig. S.2) manipulation events occurred. There were several small

fluctuations in the $CO_2$ fluxes within both treatments between the days 25 and 32. In the control, $CO_2$ fluxes were at a lower level and slowly increased until temperature was changed. Lowering temperature from 20ºC to 10ºC (Fig. S.3, Fuhrberg, day 38) drastically decreased $CO_2$ fluxes in both treatments, whereas further temperature changes had smaller effects.

The cumulative $N_2+N_2O$ fluxes were almost 8 times higher in ryegrass compared to the control treatment. $N_2+N_2O$ fluxes were initially high in both treatments (Figs. 2a and 3a) but decreased rapidly following the initial drainage period (Table

S.2). During the remainder of the experiment, fluxes remained low and were only to a minor extent affected by the experimental manipulations. Initially, the ryegrass treated cores had high $N_2+N_2O$ fluxes which rapidly decreased during the incubation. Between the first and the second (09/02 and 14/02) water content manipulation events, cores 2 and 3 responded with smaller $N_2$ and $N_2O$ (core 3 only) peaks.

The $N_2O/(N_2+N_2O)$ ratio of fluxes (Table 6) shows that $N_2O$ dominated the N fluxes. The $N_2O/(N_2+N_2O)$ ratio was similar

for both treatments. During the irrigation-fertilization period at day 31, the $N_2$ production increased in both treatments (Fig. 2b and Fig. 3b) and the $N_2O/(N_2+N_2O)$ ratio decreased (Fig. S.5). This response occurred 1-2 days after the onset of irrigation.





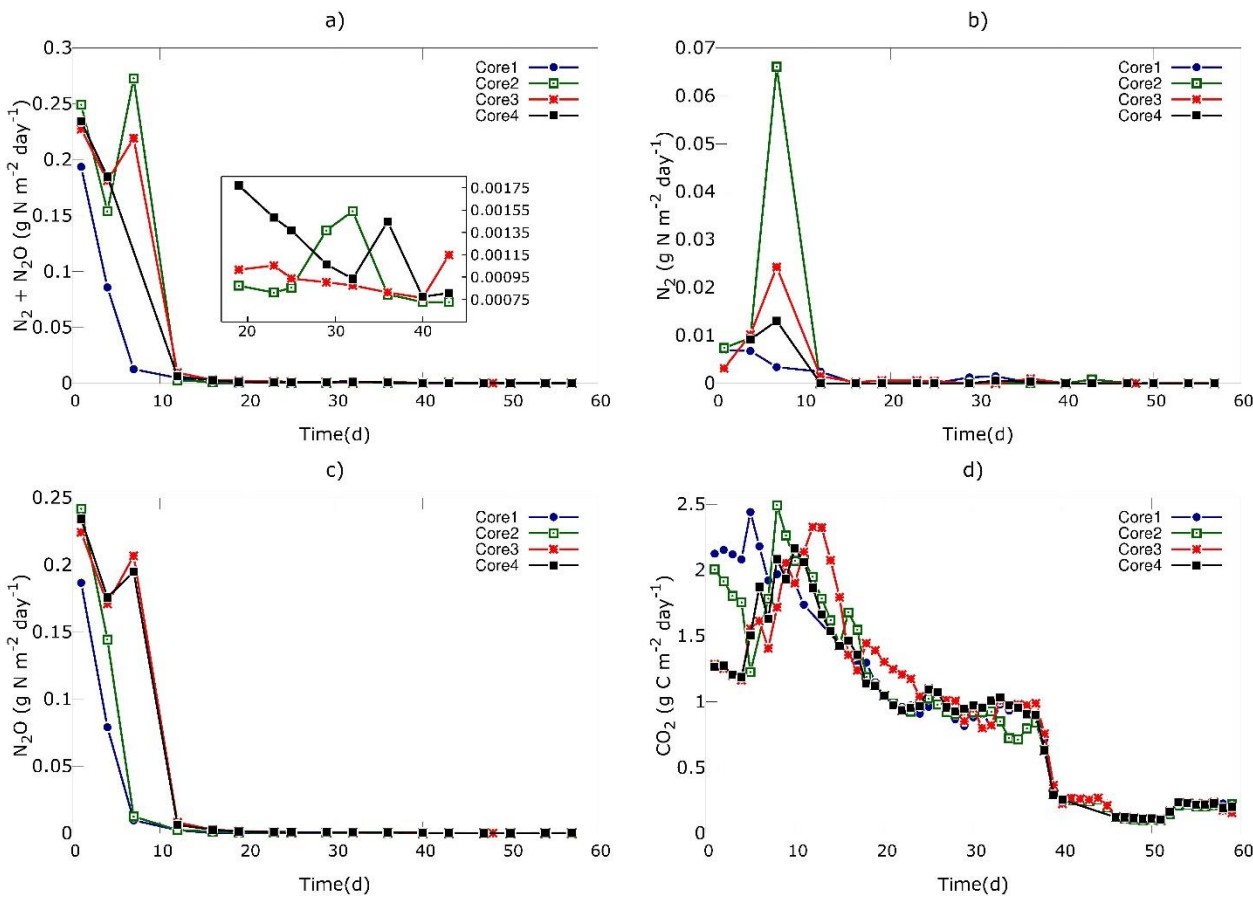

**Figure 2 a-d: Measured fluxes of (a) N₂+N₂O, (b) N₂, (c) N₂O and (d) CO₂ throughout a laboratory incubation of a sandy, arable**
**soil from Fuhrberg, Germany. The four re-packed soil cores shown were amended with ryegrass prior to incubation. The nested**
**figure in figure (a) shows the effect of the irrigation and fertilization event on Day 32.**







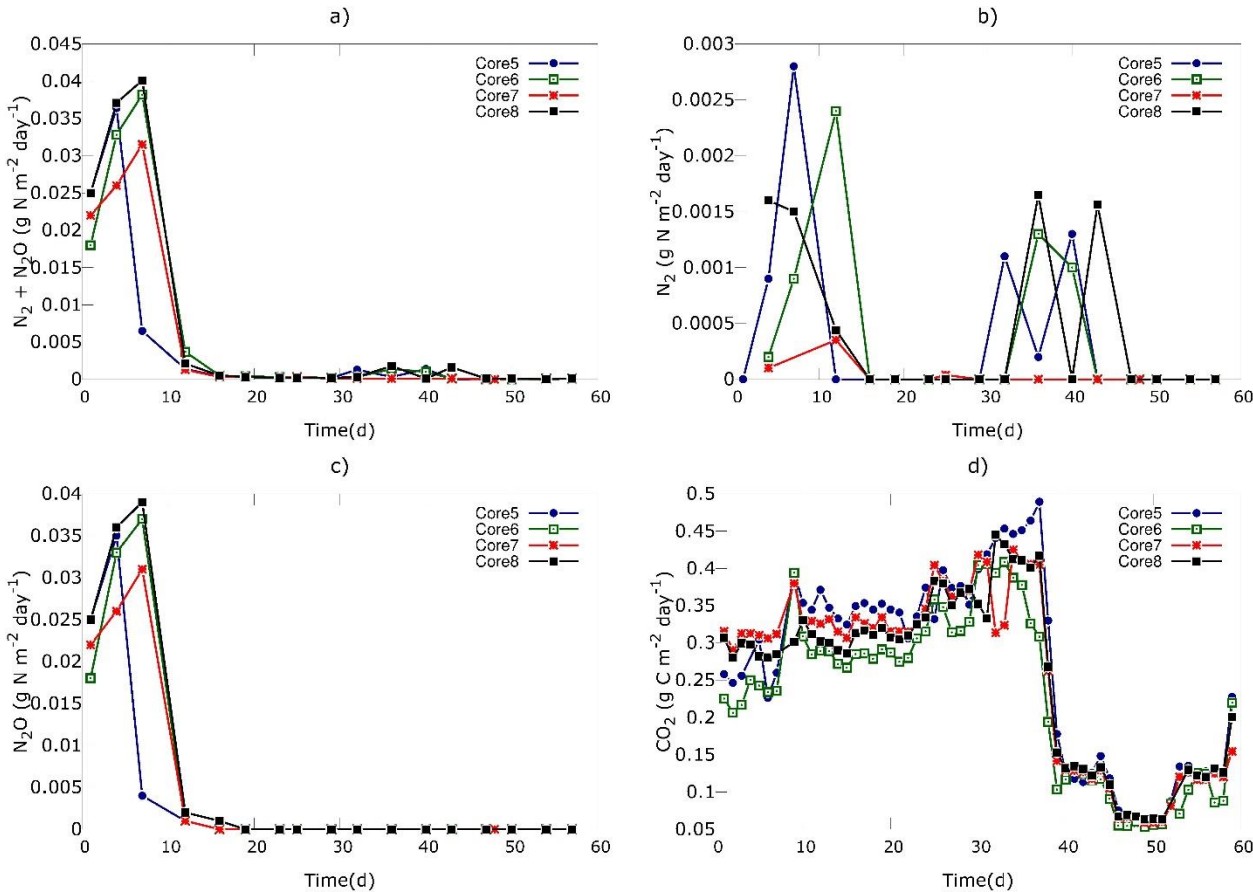

**Figure 3 a-d: Measured fluxes of (a) $N_2$+$N_2O$, (b) $N_2$, (c) $N_2O$ and (d) $CO_2$ throughout a laboratory incubation of a sandy, arable soil from Fuhrberg, Germany. The four re-packed soil cores shown had no ryegrass amendment prior to incubation.**

**3.3 Modeled results of silt-loam soil**

DeNi and Coup overestimated $CO_2$ production, with predicted $CO_2$ fluxes 3 to 10 times higher than the measured values, whereas DNDC mostly underestimated the measured fluxes (Table 8). The variability of the model calculations is quite low, and the fluctuation of the values does not always follow the changes of the measured values. The time series of the $CO_2$ flux calculation of DeNi followed the fluctuation of the temperature settings whereas the other models mostly predicted only

decreasing trends over time as shown for treatment $VI_{20N\_80\%\_1.4}$ (Fig. 4).

On average, DeNi calculated ~4 times higher $N_2$+$N_2O$ fluxes than measured. In contrast to this, $N_2$+$N_2O$ fluxes obtained from Coup were about 9 times lower than the measured values, despite the fact that the $N_2O$ estimation of Coup was quite close to the measured values. In DNDC, it is notable that $N_2$ fluxes were always zero and it therefore underestimated $N_2$+$N_2O$ fluxes even more (~30 times) than Coup. The $N_2O/(N_2$+$N_2O)$ ratio of DeNi (Fig. S.6) fitted the ratio of the





measured values quite well, whereas this was not the case for Coup, which overestimated this ratio. The time courses of the $N_2+N_2O$ fluxes of DNDC and DeNi mostly agreed with measurements but to a lesser extent for Coup (Figs. 4a-c). Coup predictions exhibited an inverse trend with measured values during the first 10 days.

Table 7: Normalized treatment effects on $N_2+N_2O$ fluxes (silt-loam soil) of modelled relative to observed results.
Treatments differ with respect to $NO_3^-$ content (10-40 mg N $kg^{-1}$ dry soil), WFPS (73-90%) and bulk density (1.4-1.52 g $cm^{-3}$). Values shown are the ratio of treatment differences between modeled and measured values, e.g. $((I_{Mod} - II_{Mod})/I_{Mod})/((I_{Meas}-II_{Meas})/I_{Meas})$.

| Coup/Measured | $II_{10N\_80\%\_1.4}$ | $III_{40N\_80\%\_1.4}$ | $IV_{20N\_80\%\_1.46}$ | $V_{20N\_88\%\_1.52}$ | $VI_{20N\_80\%\_1.4}$ | $VII_{20N\_90\%\_1.4}$ |
|---|---|---|---|---|---|---|
| $I_{20N\_73\%\_1.4}$ | -0.21 | 0 | 0 | 0 | 0 | 0.70 |
| $II_{10N\_80\%\_1.4}$ | - | -0.03 | -0.06 | -0.02 | -0.38 | -0.48 |
| $III_{40N\_80\%\_1.4}$ | - | - | 0 | 0 | 0 | 0.46 |
| $IV_{20N\_80\%\_1.46}$ | - | - | - | 0 | 0 | 0.67 |
| $V_{20N\_88\%\_1.52}$ | - | - | - | - | 0 | 0.38 |
| $VI_{20N\_80\%\_1.4}$ | - | - | - | - | - | -0.86 |

| DeNi/Measured | $II_{10N\_80\%\_1.4}$ | $III_{40N\_80\%\_1.4}$ | $IV_{20N\_80\%\_1.46}$ | $V_{20N\_88\%\_1.52}$ | $VI_{20N\_80\%\_1.4}$ | $VII_{20N\_90\%\_1.4}$ |
|---|---|---|---|---|---|---|
| $I_{20N\_73\%\_1.4}$ | -0.47 | 3.17 | 23.45 | 1.15 | -1.52 | -4.59 |
| $II_{10N\_80\%\_1.4}$ | - | 0.30 | 0.20 | 0.16 | 1.20 | 1.52 |
| $III_{40N\_80\%\_1.4}$ | - | - | 0.97 | -0.03 | 0.47 | -0.06 |
| $IV_{20N\_80\%\_1.46}$ | - | - | - | 0.32 | 0.02 | -1.25 |
| $V_{20N\_88\%\_1.52}$ | - | - | - | - | 0.39 | -0.08 |
| $VI_{20N\_80\%\_1.4}$ | - | - | - | - | - | 1.67 |

| DNDC/Measured | $II_{10N\_80\%\_1.4}$ | $III_{40N\_80\%\_1.4}$ | $IV_{20N\_80\%\_1.46}$ | $V_{20N\_88\%\_1.52}$ | $VI_{20N\_80\%\_1.4}$ | $VII_{20N\_90\%\_1.4}$ |
|---|---|---|---|---|---|---|
| $I_{20N\_73\%\_1.4}$ | -0.08 | 0.10 | 10.80 | 0.60 | -0.10 | -0.16 |
| $II_{10N\_80\%\_1.4}$ | - | 0 | 0.16 | 0.12 | 0 | 0.02 |



| | | | | | | |
|---|---|---|---|---|---|---|
| III$_{40N\_80\%\_1.4}$ | - | - | -0.99 | 1.34 | 0 | -0.02 |
| IV$_{20N\_80\%\_1.46}$ | - | - | - | 0.25 | 0.43 | 0.56 |
| V$_{20N\_88\%\_1.52}$ | - | - | - | - | 0.54 | 0.57 |
| VI$_{20N\_80\%\_1.4}$ | - | - | - | - | - | 0.04 |

After the initial 10 days and until day 29, the pattern of Coup is more or less similar to the measured values, but the magnitude of the modeled values is approximately 2-3 times smaller (Fig. 4).

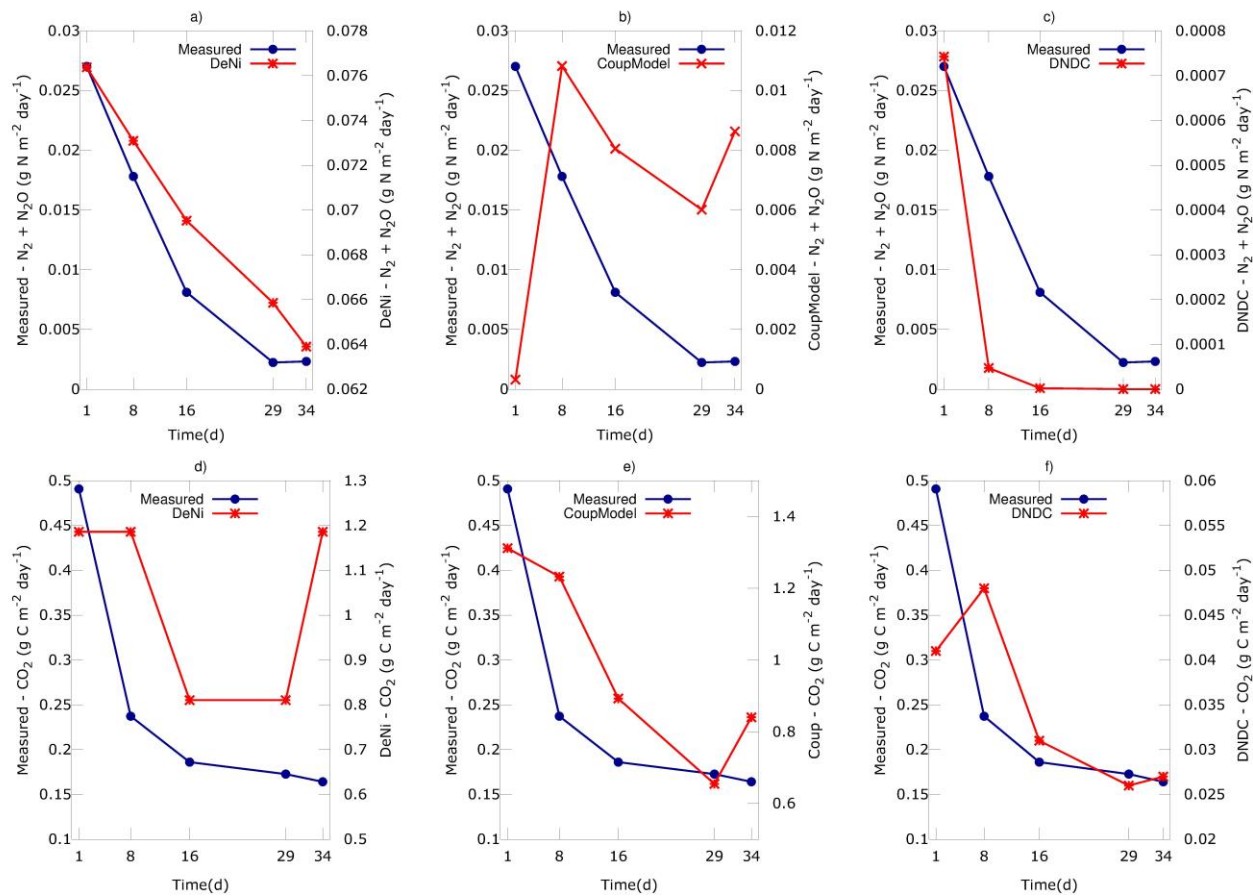

**Figure 4 a-f: An example (treatment VI$_{20N\_80\%\_1.4}$) for the measured and modeled (DeNi, Coup and DNDC) N$_2$+N$_2$O (a, b, c) and CO$_2$ (d,e,f) fluxes of a silt-loam arable soil from Hattorf, Germany**




There were a few similarities between measured and modeled fluxes when comparing the cumulative $N_2+N_2O$ fluxes (Fig. 5) and the ratio of the $N_2+N_2O$ fluxes (Table 7) of the seven silt-loam treatments. While Coup results show little variation, both measurements and DeNi exhibit a large range between minimum and maximum $N_2+N_2O$ fluxes (Fig. 5). The DeNi results follow the changes of the measured values quite well, responding to increases of $NO_3^-$ ($II_{10N\_80\%\_1.4}$ < $VI_{20N\_80\%\_1.4}$ <

$III_{40N\_80\%\_1.4}$) and WFPS ($I_{20N\_73\%\_1.4}$ < $VI_{20N\_80\%\_1.4}$ < $V_{20N\_88\%\_1.52}$ < $VII_{20N\_90\%\_1.4}$) though not bulk density ($IV_{20N\_80\%\_1.46}$ = $VI_{20N\_80\%\_1.4}$ ). In contrast, $N_2+N_2O$ fluxes by Coup increased with decreasing $NO_3^-$ ($II_{10N\_80\%\_1.4}$ with lowest fluxes). DNDC did not respond to moisture or $NO_3^-$, calculating almost the same values for all 5 treatments of the same bulk density (Table 8., $I_{20N\_73\%\_1.4}$, $II_{10N\_80\%\_1.4}$, $III_{40N\_80\%\_1.4}$, $VI_{20N\_80\%\_1.4}$ and $VII_{20N\_90\%\_1.4}$). However, DNDC responded positively to bulk density (highest values for $IV_{20N\_80\%\_1.46}$ and $V_{20N\_88\%\_1.52}$). For the change from the wettest treatment ($VII_{20N\_90\%\_1.4}$)

to each of the other 6 treatments, the Coup-estimated fluxes decreased together with the measured fluxes in 4 of 6 cases, while the DeNi-estimated fluxes increased in 4 of 6 cases (Table 7).

Normalized treatment effects of model results and measurements are shown in Table 7. The ratio between relative treatment differences of measured and modeled values is 1, if the measured and the modeled values changed with the same magnitude in the same direction. If the ratio is bigger than 1, the direction of measured and modeled values is the same, but the

magnitude of the response is bigger in the model than was seen in the measured values. If the value is between 0 and 1, the direction is the same, but the magnitude of the response is smaller in the model than was seen in the measured values. If the ratio is negative, the direction of the response is opposite in the model as compared to the measurements. For ratios of 0, there was no model response to differences between treatments.

For Coup, ratios showed that modeled treatment differences were either absent (10 of 21), lower than (4 of 21) or opposite (7

of 21) to measured differences. For DeNi, the model always responded to treatments (i.e. no 0 ratios), with most (14 of 21) cases showing a model response in the same direction as measured values, and two cases where the model had significantly higher ratios than the measured values. For DNDC, with two exceptions, ratios indicated either lower (11 of 21) or opposite (5 of 21) response of the model as compared to measured values, with 3 instances where the model did not respond (i.e. ratio of 0).


Table 8: Average measured (average of the 5 measurement events for 34 days) and modeled (Coup, DeNi and DNDC models) $N_2$, $N_2O$ (g N $m^{-2}$ $day^{-1}$) and $CO_2$ (g C $m^{-2}$ $day^{-1}$) fluxes of 7 incubation treatments for a silt-loam, arable soil from Hattorf, Germany. Treatments include different levels of $NO_3^-$ addition (10, 20 and 40 mg N $kg^{-1}$), WFPS (73-90%) and soil bulk density (1.4-1.52 g $cm^{-3}$).

| | | I | II | III | IV | V | VI | VII | SD |
|---|---|---|---|---|---|---|---|---|---|
| | | 20N_73%_1.4 | 10N_80%_1.4 | 40N_80%_1.4 | 20N_80%_1.46 | 20N_88%_1.52 | 20N_80%_1.4 | 20N_90%_1.4 | |
| $N_2$ | Meas. | 0.024 | 0.01 | 0.03 | 0.02 | 0.06 | 0.01 | 0.01 | 0.02 |





|  |  |  |  |  |  |  |  |  |  |
|---|---|---|---|---|---|---|---|---|---|
|  | Coup | 0.003 | 0.005 | 0.002 | 0.002 | 0.003 | 0.003 | 0.0022 | 0.001 |
|  | DeNi | 0.033 | 0.044 | 0.092 | 0.061 | 0.085 | 0.06 | 0.088 | 0.023 |
|  | DNDC | 0 | 0 | 0 | 0 | 0 | 0 | 0 | 0 |
| $N_2O$ | Meas. | 0.004 | 0.001 | 0.011 | 0.005 | 0.011 | 0.001 | 0.003 | 0.004 |
|  | Coup | 0.004 | 0.004 | 0.005 | 0.005 | 0.004 | 0.004 | 0.004 | 0.001 |
|  | DeNi | 0.005 | 0.007 | 0.014 | 0.01 | 0.017 | 0.01 | 0.019 | 0.006 |
|  | DNDC | 0.00075 | 0.00079 | 0.00079 | 0.00105 | 0.00142 | 0.00079 | 0.0008 | 0.00025 |
| $N_2+N_2O$ | Meas. | 0.028 | 0.009 | 0.042 | 0.028 | 0.067 | 0.011 | 0.016 | 0.02 |
|  | Coup | 0.007 | 0.009 | 0.007 | 0.007 | 0.007 | 0.007 | 0.0062 | 0.002 |
|  | DeNi | 0.038 | 0.051 | 0.106 | 0.071 | 0.102 | 0.07 | 0.107 | 0.029 |
|  | DNDC | 0.00075 | 0.00079 | 0.00079 | 0.00105 | 0.00142 | 0.00079 | 0.0008 | 0.00025 |
| $CO_2$ | Meas. | 0.324 | 0.228 | 0.074 | 0.208 | 0.032 | 0.297 | 0.038 | 0.123 |
|  | Coup | 1.033 | 0.986 | 0.986 | 0.986 | 1.033 | 0.986 | 0.795 | 0.081 |
|  | DeNi | 1.239 | 1.036 | 1.036 | 1.032 | 0.758 | 1.036 | 0.677 | 0.191 |
|  | DNDC | 0.173 | 0.173 | 0.173 | 0.188 | 0.2 | 0.173 | 0.173 | 0.011 |






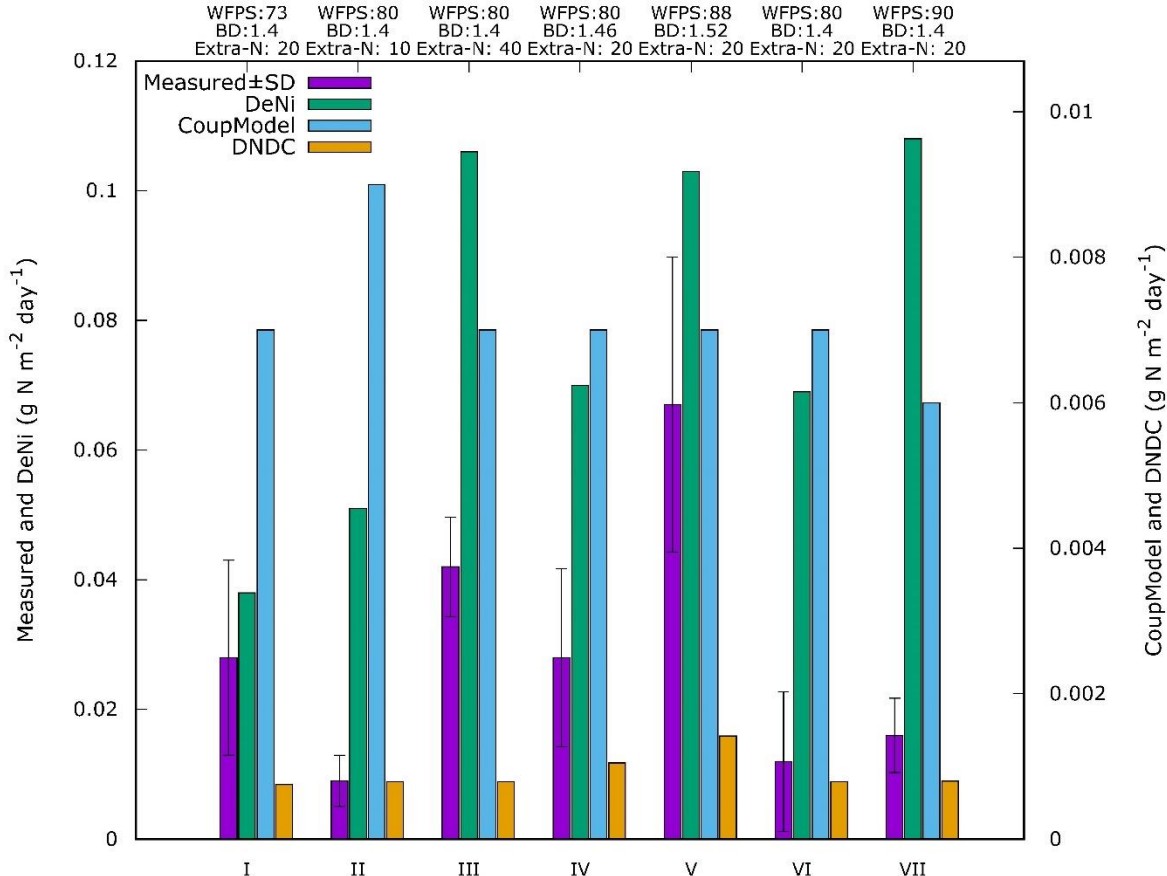

**Figure 5: Measured and modeled cumulative N₂+N₂O fluxes of a silt-loam arable soil from Hattorf, Germany.**

3.4 Modeled results of sand soil

For the ryegrass-treated sand, the Coup-estimated $CO_2$ fluxes fitted the measured emission pattern quite well (Fig. 6e). Except for an initial peak, the pattern and the magnitude of measured and modeled fluxes were almost identical. Coup overestimated the soil respiration for the control treatment (Fig. 7e), but the temporal pattern of the modeling – especially for the temperature manipulation – fitted the measured values.

DeNi overestimated the $CO_2$ fluxes for both treatments and did not respond to the labile organic C of the ryegrass treatment, since modeled $CO_2$ fluxes of both treatments were almost identical (Fig. 6d and Fig. 7d). While the pattern of the modeled fluxes followed the changes of temperature and soil water content, the magnitude of the response to these changes was too large.

DNDC calculated the smallest $CO_2$ fluxes among the three models. The modeled estimates did not reflect a litter effect and

underestimated the measured values for the ryegrass-treated soil (Fig. 6f). The model provided much better estimation for





the magnitude of $CO_2$ fluxes of the control treatment (Fig. 7f). While there was not an ideal agreement in the temporal pattern, some of the changes of the environmental conditions are clearly reflected.

Similar to the silt-loam experiment (Fig. 4b), the pattern of the estimated $N_2+N_2O$ fluxes by Coup was opposite to the trend
of the measured fluxes, exhibiting a constant initial increase in both treatments (Fig. 6b, 7b). The subsequent rapid decrease of $CO_2$ and $N_2+N_2O$ fluxes resulted from the temperature manipulation. The modeled patterns of DeNi and DNDC (Figs. 7a and c) are closer to the measured fluxes and both clearly reflect the wetting phase, which caused an increase in measured $N_2+N_2O$ fluxes of the treatment without litter but only elevated $N_2$ fluxes in the ryegrass treatment.

Comparing the order of magnitude of cumulative modeled and measured $N_2+N_2O$ fluxes (Table 9), DeNi showed agreement
in the ryegrass treatment, but overestimated fluxes of the control treatment by one order. Conversely, DNDC and Coup showed close agreement in the treatment without ryegrass but underestimated fluxes with ryegrass by one to two orders.

The $N_2O/(N_2+N_2O)$ ratio of cumulative fluxes modeled by DeNi and Coup was between 0.3 and 0.45 in both treatments (Table 9) and thus much lower than the measured ratios (>0.9, Fig. S.7, Table 9). The modeled $N_2O/(N_2+N_2O)$ ratio of DNDC was close to 1 because the $N_2$ flux estimation of DNDC was almost zero, i.e. five orders of magnitude lower than
measured fluxes.

The response of modeled $N_2+N_2O$ fluxes to increasing soil moisture following irrigation differed among models, with DeNi and DNDC predicting immediate responses (Fig. 6a and c). The response for the soil moisture manipulation of Coup was not observed during the initial growth of denitrification (Fig. 6b).



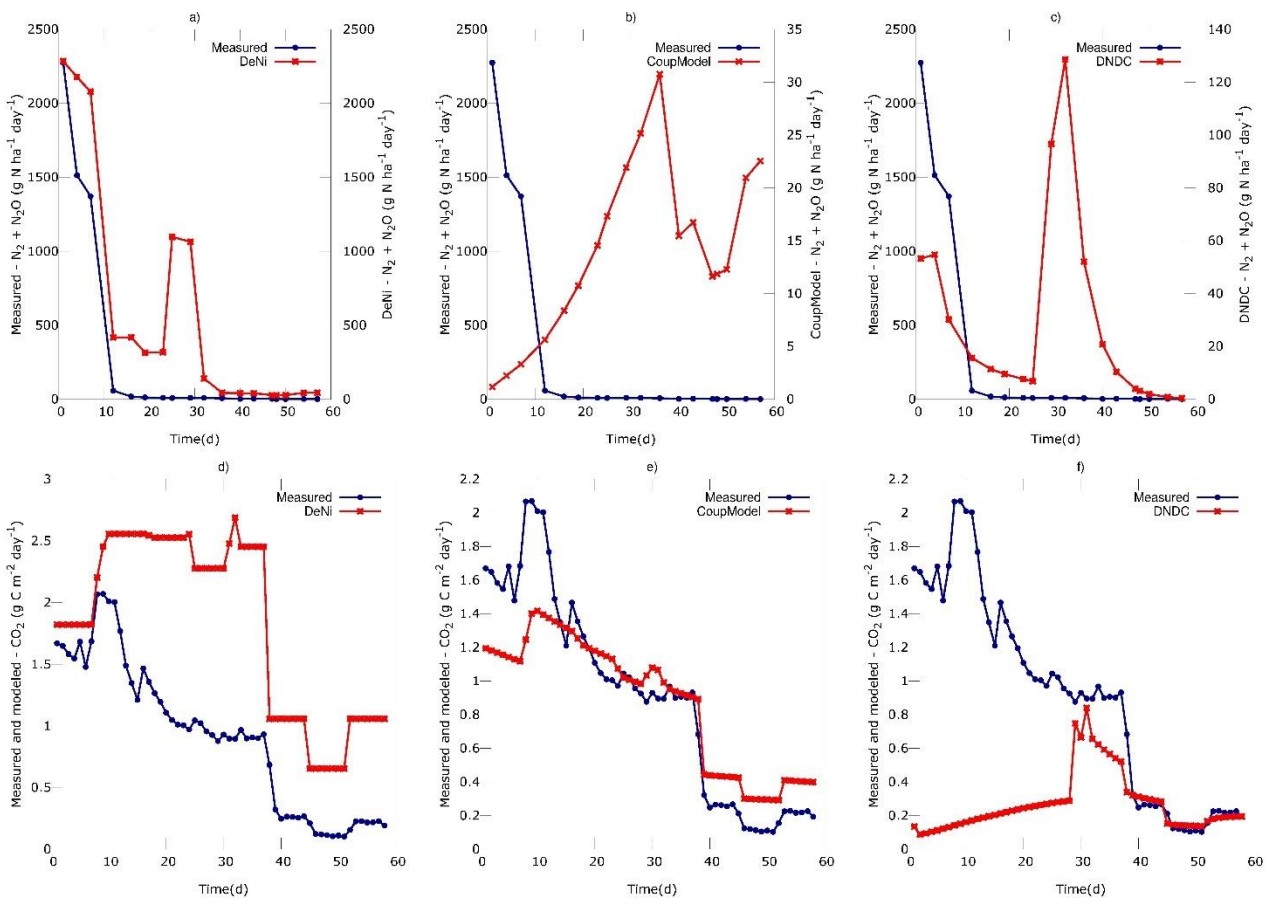

**Figure 6 a-f: Measured and modeled (DeNi and Coup) $N_2$, $N_2O$ and $CO_2$ fluxes from a 58-day laboratory incubation of soil cores from a sandy, arable site in Fuhrberg, Germany. Shown is the average of the treated cores (cores 1-4), which were amended with ryegrass prior to incubation.**

Table 9: The measured and modeled (Coup, DeNi, DNDC) average, cumulative $N_2$, $N_2O$ and $N_2+N_2O$, $CO_2$ fluxes (g N ha$^{-1}$

and kg C ha$^{-1}$) and product ratios (dimension less) for sand, arable soil from Fuhrberg, Germany. C1-4 means the first 4 parallel columns for the ryegrass treatment. The C5-8 means the 4 parallel columns of the control/non ryegrass treatment.

|  |  | Cores 1-4 (ryegrass) | Cores 5-8 (control) |
|---|---|---|---|
| $N_2O$ | Measured | 4818 | 638.5 |
|  | DeNi | 4351 | 2460 |
|  | Coup | 81.90 | 70.15 |
|  | DNDC | 507.9 | 345.4 |



| $N_2$ | Measured | 489.8 | 52.63 |
|---|---|---|---|
| | DeNi | 6264 | 4607 |
| | Coup | 170.7 | 155.8 |
| | DNDC | 0.022 | 0.019 |
| $N_2+N_2O$ | Measured | 5308 | 691.1 |
| | DeNi | 10615 | 7067 |
| | Coup | 252.6 | 226.0 |
| | DNDC | 507.9 | 345.4 |
| $N_2O/(N_2+N_2O)$ | Measured | 0.9077 | 0.924 |
| | DeNi | 0.410 | 0.348 |
| | Coup | 0.324 | 0.310 |
| | DNDC | 0.999 | 0.999 |
| $CO_2$ | Measured | 525 | 152 |
| | DeNi | 1061 | 954 |
| | Coup | 508.5 | 463 |
| | DNDC | 157.1 | 141.4 |



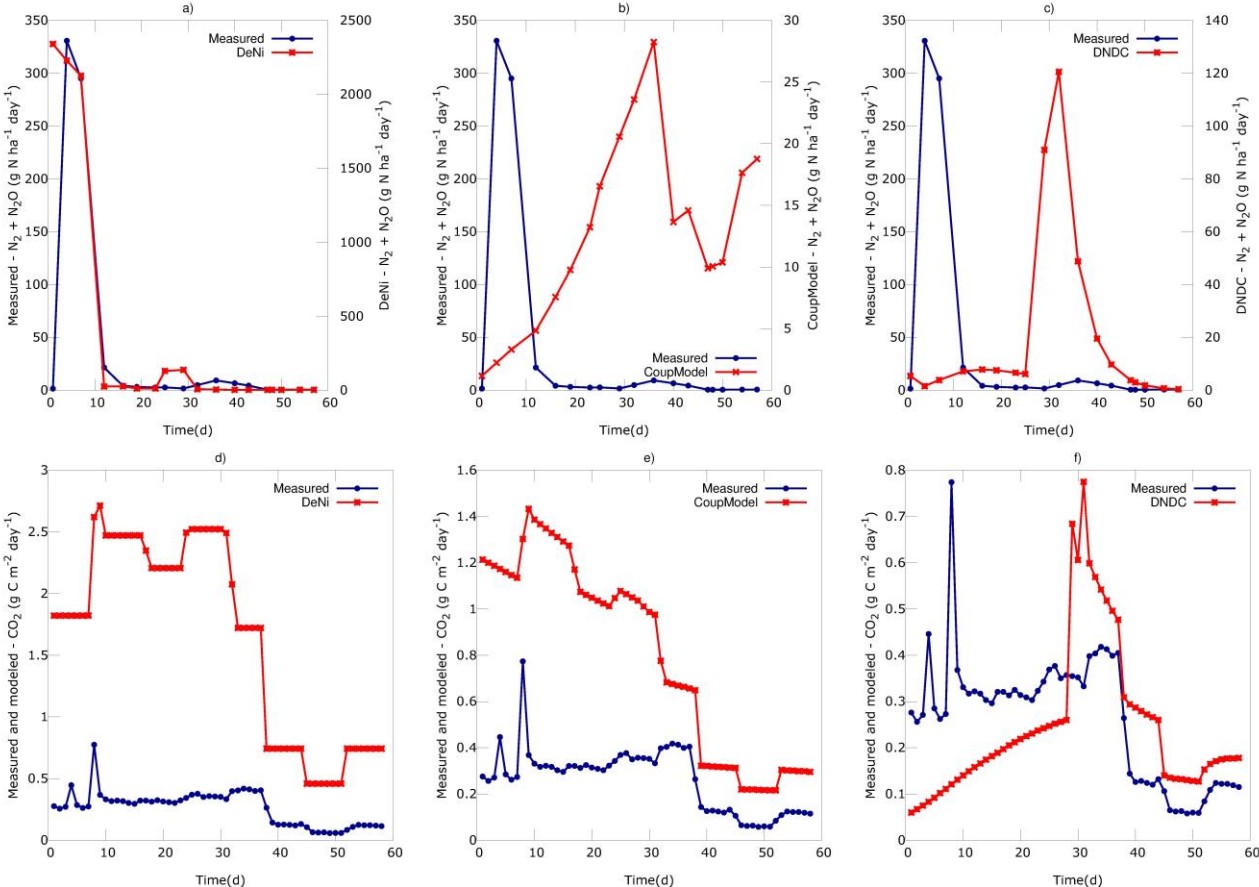

**Figure 7 a-f: Measured and modeled (DeNi, Coup and DNDC) N$_2$, N$_2$O and CO$_2$ fluxes from a 58-day laboratory incubation of soil**
**cores from a sandy, arable site in Fuhrberg, Germany. Measured values shown are the average of the control cores (cores 5-8),**
**which were given no additional substrate.**

## 4 Discussion

### 4.1 Experimental results

#### 4.1.1 Silt-loam soil

The highest cumulative CO$_2$ fluxes were measured at low WFPS/bulk density and the lowest fluxes at high WFPS/bulk
density (Table 5). Respiration thus reflected the typical responses to temperature and aeration (Davidson et al., 2000). Figure
1a shows that the total denitrification was controlled by several interacting factors, where decreasing nitrification can be
explained by the combination of substrate exhaustion and temperature (Müller & Clough, 2014). The increasing
denitrification in the wettest treatment (VII$_{20N\_90\%\_1.4}$) could be due to ongoing O$_2$ depletion resulting from respiration at low
diffusivity during the early phase of the incubation (Well et al., 2019).





The low $N_2O/(N_2+N_2O)$ product ratio (between 0.088 and 0.264, Table 5) indicated that $N_2O$ was effectively reduced to $N_2$, so that total fluxes were dominated by $N_2$. Since high $NO_3^-$ contents and low pH are known to inhibit $N_2O$ reduction (Müller & Clough, 2014), the low $N_2O/(N_2+N_2O)$ ratios might explained by near-neutral pH values or low $NO_3^-$ contents, below the reported threshold for $N_2O$ reduction inhibition (45 mg N $kg^{-1}$; Senbayram et al., 2019). The relevance of $NO_3^-$ content for

controlling the product ratio is supported by the fact that the lowest $N_2O/(N_2+N_2O)$ ratio was observed in the treatment with lowest $NO_3^-$-N concentration ($II_{10N\_80\%\_1.4}$), whereas the highest values were obtained at the highest $NO_3^-$ content ($III_{40N\_80\%\_1.4}$). However, it is notable that the highest $NO_3^-$ in this study (40 mg N $kg^{-1}$) was still below the 45 mg N $kg^{-1}$ threshold.

### 4.1.2 Sand soil


The dramatic differences between measured fluxes of control and ryegrass soils (2-4 orders of magnitude for $CO_2$ and almost 8 for $N_2+N_2O$; Table 6) can be explained by the effects of labile carbon from ryegrass on microbial respiration and enhancement of denitrification due to increased $O_2$ consumption and supply of reductants for denitrifiers (e.g. Senbayram et al., 2018). In contrast, the control soils not only had no ryegrass amendment but were also pre-incubated (decreasing the

what labile carbon was present) to avoid an initial peak in $CO_2$ fluxes after the re-wetting of the dry soil (see methods). The $CO_2$ fluxes of the ryegrass treated cores (cores 1-4) between days 4 and 12 show a rapid increase (Figs. 2d, 3d). The large response of respiration to the ryegrass treatment almost hides the smaller effects resulting from the changing water and $NO_3^-$ content, while these effects were clearly visible in the control treatment. However, small effects with a similar pattern to that seen in the control soils were also evident in the ryegrass treatments (Figs. 2d, 3d, S.4 day 25-35 increasing trend all cores

expect core 2). Other notable responses in Figs. 2d, 3d are the higher peaks of $CO_2$ on day 7 and a big decrease in the $CO_2$ flux values for both treatments on day 38. On day 7, the water content of the soil cores was decreased (Fig. S.4) and it resulted the higher $CO_2$ emission. On day 38, a simultaneous increase in water content and decrease in temperature (Fig. S.3 and S.4), which presumably caused lower $CO_2$ flux.

The time course in $N_2+N_2O$ fluxes (Figs. 2a and 3a) can be explained by the combination of easily available carbon, the

effect of soil water content and changes in the soil $NO_3^-$ content. The control treatment – without organic matter amendment – was almost one magnitude smaller than the ryegrass treated soil cores, but the initial high water and nitrate content (80% WFPS, 66 mg N $kg^{-1}$ dry soil, Table 2 and 3) resulted in higher $N_2+N_2O$ fluxes in the first 4 days of the experiment for both treatments. The water potential at the bottom of the cores was changed at day 4 and the water and $NO_3^-$ content decreased in the soil cores (Table S.1 and S.2). The increase of the water (Fig. S.4) and $NO_3^-$ content (Table S.1 at 08.03.2017) between

the days 24 and 27 led to increasing $N_2+N_2O$ fluxes in both treatments. The $NO_3^-$ content and the seepage of leachate show some variability between replicates (Table S.1 and S.2) which we attribute to the fact that initial water content (80% WFPS) was located in the steep sloping section of the water retention curve (Fig. S.2), where small changes in water potential would be related to large change in water content. The variable leaching is thus probably due to the limited precision of water





potential control (Table S.1). At 80% WFPS, our estimated uncertainty in pressure head control of 20 mbar would lead to an

uncertainty in soil water contents equivalent to 0.023 g g$^{-1}$ or 8.1% WFPS. Presumably, the possible uncertainty of the

manual compaction of the soil columns may also have resulted in minor variability in water retention properties among the

soil columns.

Seepage of the cores not only lowered water contents but also caused loss of NO$_3^-$ (Table S.1 and S.2). The high and

variability in water and NO$_3^-$ content might explain some of the measured variability in gaseous N fluxes (initially high

fluxes in both treatments, but decreasing quickly (Figs. 2a and 3a)). While the organic matter amendment clearly enhanced

denitrification in the initial phase with high water content, this was not the case during the later phases when fluxes of both

treatments were similarly low, likely since anoxic micro-sites disappeared due to improved aeration.

The product ratio of fluxes (Table 6) shows that mostly N$_2$O was emitted, which we attribute to the high NO$_3^-$-N level and

the low pH (Table 1) (Müller & Clough, 2014). The product ratio was similar with and without litter amendment. This might

indicate that the combined inhibitory effect on N$_2$O reduction by low pH and high NO$_3^-$ was more effective than the potential

enhancement in N$_2$O reduction in presence of labile C in the ryegrass treatment (Müller & Clough, 2014).

### 4.2 Possible explanations for the deviations between measurement and modeling

The goal of this work was to test and evaluate the denitrification sub-modules of the models and not to harmonize the

measured and modeled values by calibration or to rate the performance of the different models. Clearly, after calibration, the

models can simulate results of the same magnitude as the measured values. Our aim, however, was to find the missing

processes and limitations of the sub-modules for further model development.

Overall, there were large differences between the measured and modeled results. Modeled N$_2$+N$_2$O fluxes were between 10

and 580% of measured fluxes in the silt-loam incubation and between 1 and 9060% in the sand incubation (Table 8 and 9).

DNDC, originally developed to accommodate field conditions, calculated almost zero N$_2$ emissions for both treatments. The

structure of the model with a simple soil water management sub-module, rather than the option to manually set up the daily

soil water content, may not be a good fit for laboratory experiments. The model provided a higher amount of leachate in the

first days of the simulations. This could be the reason for the lower N$_2$O and the almost zero N$_2$ production.

In theory, there should be a certain lag time between rainfall or irrigation and the occurrence of denitrification in the soil

(Tiedje 1978; Smith and Tiedje 1979). DNDC ignores this lag time (Fig. 6c and 7c, day 25), as shown by the modeled N$_2$

and N$_2$O fluxes, which occurred almost immediately after the rewetting of the soil. In contrast, because Coup assumes

growth of denitrifiers as a prerequisite of denitrification, there are no abrupt changes in the modeled denitrification, as any

possible response was masked by the ongoing growth of the denitrifier community (refer to 4.2.3).

There was also disagreement in the N$_2$O/(N$_2$+N$_2$O) ratio and the temporal dynamics of the modeled fluxes, which did not

always fit well with the measurements (Fig. 4, 6, 7 (a, b, c), S.6 and S.7). Models were used with the default settings of

coefficients because it was not the objective of this study to calibrate them using the measured data. It was therefore not



expected that the modeled data would fit the magnitude of measured fluxes. However, the poor fit in temporal dynamics and the $N_2O/(N_2+N_2O)$ ratio shows that some of the model routines were not adequate to obtain correct responses to the denitrification control parameters established in the experiments. It is notable, though, that some model parameters were not

assessed in the experiments (e.g. labile C content, denitrifier biomass, anaerobicity of the soil) and also that the temporal and spatial resolution in the measurement of control factors such as mineral N and soil moisture was limited; including these may have improved model estimates

### 4.2.1 Complexity of models

The agreement of measured and modeled results depends not only on the experimental set-up, and to which extent model parameters are represented by measurements, but also on the model complexity. DNDC and Coup are complex, with more parameters and more elaborate descriptions of denitrification and decomposition than DeNi. However, using this detailed approach may allow some factors to dominate the denitrification calculations and give biased results (Metzger et al., 2016). Laboratory mesocosm experiments simplify 'real' field conditions, and the simplicity of DeNi could be one reason why it

had reasonably good success modeling the incubation experiment. The pure nitrification and denitrification approach of DeNi minimizes the influence of sub-modules that represent more complex processes, which are present in Coup and DNDC. For example, rather than using a water management sub-model, we were able to input measured daily water and soil $NO_3^-$ content into DeNi, which may have contributed to the better predictions. In contrast, Coup and DNDC use sub-models to predict changes in soil water and $NO_3^-$ content. Coup has an option to overwrite the calculated daily water content data,

which we used, but this option was not available for DNDC. In fact, Coup provides numerous options to turn on or off different sub-modules and use constant values instead of dynamically changed parameters (Table S.6 and S.7). Simplifying by turning off sub-modules decreases the complexity of the model and, in 'simplified' experiments, such as ours, may actually improve the final results.

### 570 4.2.2 Labile organic carbon (litter)

While treatments without litter amendment were relatively low in labile C content, the ryegrass treatment was established to mimic incorporation of crop residues and thus contained large amounts of labile organic C. Coup and DNDC provide options to modify the labile C and N pools, and in running these models, the C and N content of the ryegrass was added to the respective labile pools. However, the results of Coup and DNDC didn't reflect the extremely fast decomposition that was

observed in the experimental results (Fig. 6f). Although the measured results showed that soils with ryegrass amendment had 345% higher $CO_2$ than control soils (Table 9.), Coup calculated similar $CO_2$ fluxes for both treatments (Fig. 6e and 7e) as well as only a 10% difference in the modeled $N_2$ and the $N_2O$ fluxes between the treatments (Fig. 6b and 7b). DNDC actually calculated 40% higher $CO_2$ fluxes for the control treatment as compared to the ryegrass-amended soils (Fig. 6f and

7f). The decomposition rate of the ryegrass in the models needed to be much higher than the decomposition rate that is
currently provided for the labile pools. Because DeNi has a simple, one C pool approach for calculating soil respiration, it
was also unable to handle the extra ryegrass as rapidly decomposable carbon. Similar to Coup, DeNi calculated similar soil
respiration and $N_2+N_2O$ fluxes for both treatments of the sand soil (Fig. 6a,d and 7a,d).

In these models, decomposition processes are assumed to be driven by soil water content and temperature (Table S.6), thus
the microbial response to treatments (e.g. $NO_3^-$ addition, pH), although they are known to influence microbial carbon use
(Manzoni et al., 2012), cannot be explicitly simulated. It should also be noted that decomposition of the labile and
recalcitrant pools in these models are calculated independently. However, field and empirical data (Kuzyakov, 2012) suggest
adding labile C could also enhance the decomposition of resistant pool, e.g. priming effects, which none of these models
account for. Our results also suggest the importance of simulating microbial dynamics of decomposition explicitly to better
account for the drivers of decomposition, because these ultimately influence the denitrification flux estimations. It means
that the direct application of these models with first order kinetics for decomposition to simulate the effects of fertilization or
changing N deposition on denitrification fluxes could be largely biased. Future research should aim to quantify more
appropriate decomposition rates for models to better take into account labile pools.

### 4.2.3 Denitrifiers

In Coup, the biomass of denitrifiers directly limits the maximum denitrification rate. We assume that the slow increase of
fluxes obtained from Coup (Fig. 6b, 7b) was due to the modeled growth of denitrifiers, since the default setting assumed a
low abundance of denitrifiers, hence the denitrifiers had to first grow before reaching maximum denitrification rates
(denitrifier growth was observed in the model output although this data was not shown). It can be concluded that when
modeling denitrification during incubation experiments, the model initialization must include inducement of denitrifier
growth.

Another reason for the slow increase of the denitrifier biomass at sandy soil modelling could be that the modeled anaerobic
soil volume fraction (ansvf) is orders of magnitude smaller than the measured ansvf (Rohe et al., 2020) and the small ansvf
was not ideal for the growth of denitrifiers. This may have led to a non-realistic, too small denitrifier community, and
therefore low $N_2O$ and $N_2$ fluxes.


### 4.2.4 Anaerobic soil volume fraction

DNDC and Coup use a similar calculation of the ansvf and both models use it for the calculation of the denitrification
processes. While the ansvf estimations of DNDC were not available as an output, the Coup results were obtained showing
that ansvf was almost constant. This is not plausible since the parameters affecting ansvf, i.e. diffusivity and $O_2$
consumption, must have been highly variable since soil moisture and respiration exhibited large differences between





treatments and experimental phases. The underestimation of $N_2+N_2O$ fluxes by Coup could therefore result from the inappropriate calculation of ansvf in the model (see in section 4.2.3).

One of the main goals of this study was to test the ability of the existing biogeochemical models to predict the temporal dynamics of $N_2$ and $N_2O$ fluxes and identify where the models could be improved. Ensuring correct ansvf calculations could

significantly improve the efficiency of denitrification sub-modules, and thus further work on these algorithms within Coup is one area for future research that we would strongly recommend. Similarly, it would be beneficial to test the ansvf calculations of DNDC, which was not possible in our study, as the source code was not available and the ansvf is not included in output data.

**4.2.5 Determination of control factors in the experiments**

Within the sand incubation, another reason for the underestimations of denitrification products by Coup and DNDC could be properties of the soil itself. The soil had a low pH, which has a direct influence on denitrification processes (Leffelaar and Wessel, 1988). However, while the denitrification sub-module of DeNi is sensitive to changes in soil temperature, moisture, $NO_3^-$ and SOC content, the pH of the soil only influences nitrification processes. Therefore, the low pH may have had less

effect on the $N_2O$ flux estimation of DeNi, as compared to Coup and DNDC. Another reason for the smaller denitrification fluxes of Coup and DNDC could be the soil texture. Texture influences the hydrology, the anaerobic soil volume fraction and the diffusion of the gasses, which altogether control denitrification processes. According to the water retention curve, the range of water contents in the incubation were located in a section of the curve where small changes in water potential could lead to large changes in WFPS (Fig. S.4). In Coup and DNDC, WFPS has multiple effects on denitrification through

respiration and diffusion processes. The challenge for these models is to describe these direct and indirect effects correctly to match the observed response of denitrification. Because DeNi does not use a fully process-based approach, the effects of environmental factors – like WFPS – are considered with various empirical functions. We suspect that the use of empirical functions (functions derived from experimental lab data to describe WFPS) was more successful in modeling WFPS effects on denitrification than the fully process-based approaches.


**4.3 Suggestions for future model evaluation experiments and model improvement**

This study has demonstrated advantages and shortcomings of modelling denitrification processes using current models. We suggest the following to improve model algorithms and parameters by targeted experimental studies: (1) design experiments to specifically evaluate sensitive input variables (e.g. decomposition of labile organic carbon), which can then be used to

improve current model algorithms; (2) take more frequent measurements in future studies (ideally daily) to allow better descriptions and evaluations of temporal dynamics (3) use updated techniques to take measurements. To the best of our knowledge, all previous model evaluation studies (NGAS and DailyDayCent) using measured denitrification data were





based on the outdated acetylene inhibition technique (Bollmann and Conrad, 1997; Nadeem et al., 2013; Sgouridis et al., 2016). Future studies should (as was done in this study) be based on He/O$_2$ or $^{15}$N gas flux methods; (4) take measurements to evaluate unknown/hypothetical parameters in the model equations (e.g. static growth and death rate of the denitrifiers, rate coefficients for the different denitrification processes, etc.) (5) adapt models so that the parameters better represent measurements from real soils (e.g. measured SOC fractions v.s. SOC pools in the models); (6) continue to re-evaluate how processes are describe in models. These models were developed decades ago, and new technical solutions appear constantly. There are several missing or poorly described processes in the models. Strong simplification of some process descriptions (e.g. no or inadequate or poorly calibrated microbial dynamics (see section 4.2.2)) have to be overcome or their implications have to be estimated. Further experiments are thus necessary to describe more precisely the effect of temperature, moisture and substrate manipulations on the microbial/denitrifier community and therefore on N$_2$ and trace gas fluxes. These kinds of datasets will help to (i) identify inadequate process descriptions, (ii) calibrate the sub-modules separately from the other parts of the model and finally, (iii) develop new, better approaches for the description of the processes.

## 5 Conclusions

The goal of this study was to assess the ability of the denitrification sub-modules in three biogeochemical models to predict the N$_2$ and N$_2$O fluxes of incubated soils in response to different initial soil conditions and changing environmental factors. The results show that the models did not calculate fluxes of the same magnitude as the experimental results; measurements were overestimated by DeNi and underestimated by DNDC and Coup. However, with only a few exceptions, the temporal patterns of the measured and modeled emissions were quite similar for the sandy soil. For the silt-loam soil, Coup and DNDC showed no response in 47% and 14% of cases, respectively, and responded in the same direction in 19% and 52% of cases, respectively. For DeNi, the model responded in the same direction in 67% of cases, with 33% in the opposite direction. Treatment responses of the models suggest that in addition to calibration, improvement of the model functions is needed to better predict N$_2$O and N$_2$ fluxes from denitrification. While none of the models was able to determine litter-induced decomposition dynamics correctly, the complex models Coup and DNDC were apparently further hampered by their limited ability to give realistic estimates of soil moisture, anaerobic soil volume and denitrifier biomass. The simple structure of the DeNi model, using more empirical functions, thus can be more accurate for some experiments. This suggests that the potential advantage of Coup and DNDC to include more control factors is only useful when the control factors have been more thoroughly researched and respective functions are more reliable. Developing reliable functions for complex control factors requires experimental data with more detail in temporal resolution and parameter determination. Further development of the models to overcome the identified limitations based on experiments with enhanced denitrification activity and control parameter determination can largely improve the predicting power of the models.



**Acknowledgments**

This study was funded by the Deutsche Forschungsgemeinschaft through the research unit DFG-FOR 2337: Denitrification
in Agricultural Soils: Integrated Control and Modelling at Various Scales (DASIM). We thank the laboratory teams of
Institute of Soil Science and Centre for Stable Isotope Research and Analysis of Göttingen University and Thünen Institute
of Climate-Smart Agriculture, Braunschweig, for support in analysis and experiments regarding the Hattorf and Fuhrberg
studies, respectively. Specifically, we thank Marrtina Heuer, Ute Rieß, Jennifer Giere, Reinhard Langel and Lars Szcwec for
isotopic analyses, Ute Tambor, Sabine Watsack, Karin Schmidt and Ingrid Ostermeier for further analysis and sampling and
Stefan Burkart for supporting automated incubations. We further thank Christine von Buttlar, Ingenieurbüro Landwirtschaft
und Umwelt (IGLU), Göttingen, for supporting the work at the Hattorf site.

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
