# Peer review of "Evaluation of denitrification and decomposition from three biogeochemical models using laboratory measurements of N2, N2O and CO2"

_Biogeosciences, 2021_

## Author Response (AR1)

We would like to thank Reviewer 3 for the time and effort that they took to provide feedback for our manuscript. Their efforts towards improving our manuscript were much appreciated. We have made significant modifications reflecting those comments, which are outlined below.

Reviewer 3:

The manuscript "Evaluation of denitrification from three biogeochemical models using laboratory measurements of N2, N2O and CO2" describes a model evaluation approach and a lab experiment for N2O emissions. The lab experiment is set up for two different soils from Lower Saxony in Germany and includes variations of soil water content by continuous measurements of N2, N2O and CO2 fluxes. These experimental results are used for model evaluation on these variables.

The study touches is crucial topic. The demand for these kind of studies is well indicated and conclusive. However, there are a couple of concerns about how the study is applied and presented.

Q1 and Q2: To which extend do the lab experiments help to evaluate and improve the models? The fluxes are highly sensitive to soil water content and by sieving and homogenization of the soils, the structure is affected. Annual sums of GHG fluxes are often driven by the most extreme impacts (e.g. tillage, fertilizer application, rainfall impact), which is not included in the evaluation. The study focusses on low temperatures and varies the soil water content. The chance with controlled conditions is to cover a wider range. Obviously, studies always have limitations, however, some of these aspects could be discussed in the manuscript. The lack of connection to field experiments (not in terms of measurements, but as discussion point) makes it difficult to follow the argumentation.

I was surprised to read that the change from 1.4 to 1.52 g/cm3 bulk density is tested. These models are tested for field application and possible spatial applications. Assuming that bulk density is estimated by a general assumption, these differences are close to the range of uncertainty. I acknowledge and appreciate that the authors set the bulk density to the values measure in the field. However, as these are set up in the lab anyway, I wonder why not more extreme steps were tested, to get some real differences and include the bulk density as additional factor in the analysis.

**R1 and R2: Models are often tested on extreme conditions, and we assumed that these models would respond to those appropriately. We deliberately chose small changes to see whether the models could respond to finer changes. While the denitrification sub-models were developed based on laboratory experiments, their test an improvement are also obvious with laboratory experiments. Important to note, that the difficulties of the $N_2$ measurements don't allowed us to calibrate or improve the models with $N_2$ flux data from**

the fields. We must use for these calibrations and improvements the results of laboratory measurements.

To clarify the need to test models under controlled conditions with measured $N_2$ and $N_2O$ fluxes we added the following section.

L. 67/74: "While in many studies $N_2O$ emissions alone are used to develop and train models (Chen et al., 2008), measurements of both $N_2O$ and $N_2$ fluxes are necessary to develop and/or test algorithms (Leffelaar and Wessel, 1988; Parton et al., 1996; Del Grosso et al., 2000). "

L. 74/76: "However, although targeted experiments often focus on large differences in control factors (Li et al., 1996; Jiang et al., 2021) datasets of small, field-relevant changes in control factors are also necessary in order to validate models and improve their accuracy with respect to denitrification."

Q3: I think it is an excellent idea to apply the models without calibration, as this shows the impacts. However, the "default" parameters have already a history and might be by chance closer or further away to/from the optimum setting for this approach. Therefore, I think an additional run, with calibrated models show the potential of the models. It might show that the structure of the model does not allow the correct simulation of these experiments (time step, micro-scale, tipping bucket approach for soil water content, etc.).

R3: We agree with the Reviewer that after the calibration of the models the magnitude of the calculated fluxes would be much closer to the measured. However, it was not the goal of our work. We also wanted to show the potential of the models but from a different aspect. We compared the temporal dynamic of the measured and modeled fluxes to find the limitations and the missing processes of the sub-modules. The similar magnitude of the measured and modeled fluxes was just a marginal aspect. Adding additional content, such as an additional run, with calibrated models, would be interesting, but not directly relevant to our objectives, and as all of the reviewers agreed that the length of the article was problematic, we need to focus the paper strictly on the information that relates to our objectives.

L. 456/460: "Overall, there were large differences between the measured and modeled results. A clear possibility for some deviations between measurement and modeling is our choice not to calibrate the models. Clearly, after calibration, the models should better simulate our measurements. Our aim, however, was to find the missing processes and limitations of the sub-modules for further model development, rather than to harmonize the measured and modeled values by calibration"

Q4: The models are not designed for lab experiments and the partly empirical models and parameters are derived in the field, not in the lab. The authors acknowledge this (lines 535-

537). However, if this is the case, do not write the publication. If you write the publication, work out the benefits and gains you get from the study.

**R4: We don't fully agree with the reviewer here. In fact, while one can argue that denitrification sub-modules are not *designed for* lab-experiments (in the sense that their ultimate purpose is for use in modelling field activity), laboratory experiments were used for the development of all three denitrification sub-modules (DNDC, CoupModel (denitrification sub-module of PnET-N-DNDC), DeNi (denitrification sub-module of NGAS/DailyDayCent)). For example, if we read the manual of DNDC, we can find the following text on page 6: "Classical laws of physics, chemistry and biology, as well as empirical equations generated from laboratory studies, have been incorporated in the model to parametrize each specific geochemical or biochemical reaction."**

**L. 71/76: "While models are intended for use in the field, and ultimately the goal is for them to be accurate under field conditions, in order to describe processes accurately, it is often necessary to test and develop the sub-modules under controlled conditions, using targeted laboratory experiments (i.e. DNDC Scientific Basis and Processes, 2017).**

**L. 81/86: "Within each of the three models, the denitrification sub-modules use different approaches to address the complexity of denitrification, including how they consider controlling factors (e.g. soil moisture, heat transfer, nitrification, decomposition, growth/death of the denitrifiers) as well as how they simulate temporal and spatial dynamics. However, to our knowledge evaluation of the denitrification sub-modules of these models was limited due to the lack of proper N2 datasets. There is a difficulty measuring the N2 flux in the field and the very few laboratory experiments ($^{15}$N or He/O$_2$ gas flux method) are so far the only option to validate N2 fluxes and use the data for model evaluation.**

Q5.1: mection 4.2.1 is more or less a reviewer section for rejecting the manuscript.

**R5.1: What the reviewer interpreted here as a reason for rejecting the manuscript (our discussion of how model complexity interfered with model setup of our data), was perhaps poorly explained. This is, in fact, a really important point that we wanted to make about model structure. The issue here is not that models were too complex to model our data, specifically (presumably what the reviewer understood us to be saying), but that they are not designed to deal properly with datasets that are not perfect. The complexity of the models is an excellent option to have, but they need to be flexible enough to deal with real-world datasets, especially something as important as moisture dynamics (see discussion section 4.2.2).**

**We have changed this section to now say:**

**"DNDC and Coup are complex, with more parameters and more elaborate descriptions of denitrification and decomposition than DeNi. However, using a detailed approach may allow some factors to dominate the denitrification calculations and give biased results (Metzger et al., 2016). For example, the almost-zero N$_2$ emissions that DNDC estimated for**

both experiments may be reflecting how soil water is managed in the model. There is no option to manually enter daily soil water content, and the soil water management sub-module has been shown to be problematic (Smith et al., 2008; Smith et al., 2019; He et al., 2019, 2018; Brill et al., 2017; Congreves et al., 2016; Dutta et al., 2016a; Cui et al., 2014; Abdalla et al., 2011; Uzoma et al., 2015; Deng et al., 2011). The DNDC model estimates of water in this study resulted in too much leachate in the first days of the simulations (data not shown) and could be the reason for the lower $N_2O$ and the almost zero $N_2$ production."

"For DeNi, we were able to input measured daily water and soil $NO_3^-$ content, which allowed those values to be more accurate than model estimates. Coup does have an option to overwrite the calculated daily water, which we used, but this option was not available for DNDC.  The option to turn off sub-modules decreases the complexity of models in situations where that added complexity is not relevant or even problematic, as in the case of soil water mentioned above."

Q 5.2 A clear target of the paper is missing.

**R5.2: We agree that the targets of the paper were not clear enough - this is a major point from all of the reviewers, which we are addressing in our revision as follows.**

**L. 93/101: "In this study we aim to identify missing processes or limitations in the denitrification and decomposition sub-modules. We use newly measured data to test the sub-modules of existing biogeochemical models under field-relevant ranges in control factors. No systematic calibration of the model parameters was conducted since our intention was to evaluate the general model structure or 'default' model runs. Without calibration, we can compare the performance of the sub-modules with the same (factory) settings for the different experimental treatments. Specifically, our aims were to: (i) compile and present unpublished $N_2$, $N_2O$ and $CO_2$ results from two laboratory incubations (Ziehmer, 2006, Merl, 2018) (ii) simulate denitrification and decomposition using the three models (Coup, DNDC, DeNi) (iii) compare the measured and modeled temporal dynamics, (iv) make suggestions for model improvement."**

Q6: What is the new and relevant information that can be drawn out of this study?

**R6: As we've stated before, our goal here is to contribute to the development of three models in order to improve the accuracy of denitrification-related modelling. This comes with two 'new' sets of information: (1) as-yet unpublished data of $N_2$ and $N_2O$ emissions, which are sorely lacking in denitrification research and (2) using this dataset, a thorough test of the modelling capabilities of the denitrification and decomposition sub-modules of 3 models, using field-relevant changes in control factors**

**L. 98/101: "Specifically, our aims were to: (i) compile and present unpublished N2, N2O and CO2 results from two laboratory incubations (Ziehmer, 2006, Merl, 2018) (ii) simulate denitrification and decomposition using the three models (Coup, DNDC, DeNi) (iii)**

**compare the measured and modeled temporal dynamics, (iv) make suggestions for model improvement.”**

Q7: The advantage of the experiment is that all parameters are controlled and there is low variability over time. Additionally, the model structure is well known. This advantage is partly used in the discussion, but not always correctly and missing out on some aspects. I mentioned already, the relation to real field data would be great (how much higher or lower are the fluxes). Overall, the presentation of the study should be more positive.

**R7: We agree that the model structure is well known but the influence of the model structure, for example on the denitrification sub-modules not. For example, as a model developer, I found it interesting to see how the complexity of the models affected the calculated results. It is unclear exactly what the reviewer would have liked to have seen (i.e. what was incorrect/missing), but the discussion is one aspect of the paper that we have significantly changed in response to other comments, so we hope the improvements in clarity address this concern.**

**We agree that it would be interesting to be able to compare modeled denitrification estimates to field data, but that would be a different study. Our focus was testing for specific areas of improvement in how we describe denitrification and decomposition in models. But field conditions, with intact, heterogeneous soil, could not provide the proper, controlled conditions for that investigation.**

Q8: The reported error is very high, because the fluxes are very low. Considering the uncertainty of the models (discussion missing) small differences would be acceptable. The question about how these small error would affect simulation results over a year in the field is not answered (see above).

**R8: Yes, fluxes were relatively low as expected; field $N_2O$ emissions are often even lover than our $N_2O$ flux results (Rees et al., Nitrous oxide emissions from European agriculture – an analysis of variability and drivers of emissions from field experiments, Biogeosciences, 10, 2671–2682, https://doi.org/10.5194/bg-10-2671-2013, 2013.)**

**Increasing complexity of models can result in increased structural uncertainties that emphasize the incomplete knowledge of the described underlying approaches and process. Structural uncertainty analysis would help to find the knowledge gaps and the goal of our work is similar, but we used a different method to find the knowledge gaps. Rather than focusing on magnitudes, we used the temporal dynamic of the $N_2+N_2O$ and $CO_2$ fluxes to identify missing or inadequately described processes.**

**The duration of our experiments was 34 and 58 days. This was enough time to assess how the models responded to the small changes that we were testing, but it would be inappropriate to use those results to assess potential differences in annual estimates. However, it is a key point to note, that our goal is to help the models move beyond just**

being accurate at the annual scale, and instead provide accurate descriptions on a smaller time scale. Whether or not an improvement in small-scale accuracy makes a significant change to annual estimates would be a next step to test, once improvements and developments are made based on what we found in this study.

Q9: At least some statistics are required to show at least to indicate the variance/standard deviation of the replicates for the different measurements.

**R9: This is a really good point, and we have added SD values to Table 4 (L.: 286)**

Q10: The models are designed for filed experiments and simplify processes. Therefore, they are excellent over longer time periods and can be applied on different conditions. The mdoels are not designed for short term and very controlled conditions. Applying a large amount of fresh carbon is a system, only the fast pool is relevant. It doesn't matter how many pools are in the model, as the turnover time of the fast pool should be controlling the fluxes. Considering the results, the conclusion is that the other (slower) pools contribute to the emissions (if measure fluxes are over-estimated) or occupy SOC that does not contribute to emissions (if measurements are under-estimated). This can be analysed.

Q10.1: The models are designed for filed experiments and simplify processes.

**R10.1: Please read answer above (R4)**

Q10.2: The mdoels are not designed for short term and very controlled conditions.

**R10.2: As we mentioned above, we agree with the reviewer that the models have until now mostly been used in the field to assess longer time periods, but the short time-period modeling and comparison of the temporal pattern of the measured and modeled fluxes is also part of model evaluations (for instance: Li et al., 1992b. A model of nitrous oxide evolution from soil driven by rainfall events: 2. Model applications. Journal of Geophysical Research 97:9777-9783.). We, of course, hope to increase confidence in model accuracy over shorter time periods with our suggested improvements. However, we would not agree that the sub-modules cannot currently be used (or tested) using short term or very controlled conditions. It is important to distinguish between what the models were designed for (the goal was to estimate activity in the field), and what they were designed with. A mathematical description of reality (physical, chemical and microbiological processes) is always going to require simplification, and the data to establish these mathematical correlations were mostly produced by highly controlled laboratory experiments. Given that they were developed in lab experiments, it is also appropriate to test and improve the sub-modules based on lab experiments.**

**Please, let us cite from the article "Li et al., 1992a. A model of nitrous oxide evolution from soil driven by rainfall events: 1. Model structure and sensitivity. Journal of Geophysical Research 97:9759-9776.":**

"Soil thermal-hydraulic flux, aerobic decomposition, and denitrification submodels of DNDC (DeNitrification and DeComposition) work together in simulating $N_2O$ and $N_2$ emissions with a 1-day time step (1 hour during rain events)."

We have written the following evaluation of the denitrification sub-module of DNDC related to short temporal response to the rainfall:

L. 496/499: "Another issue with DNDC is response time. In theory, there should be a certain lag time between rainfall or irrigation and the occurrence of denitrification in the soil (Tiedje 1978; Smith and Tiedje 1979). DNDC ignores this lag time (Fig. 2c and 3c, day 25), and modeled $N_2$ and $N_2O$ fluxes instead occurred almost immediately after the rewetting of the soil."

Q10.3: Applying a large amount of fresh carbon is a system, only the fast pool is relevant. It doesn't matter how many pools are in the model, as the turnover time of the fast pool should be controlling the fluxes. Considering the results, the conclusion is that the other (slower) pools contribute to the emissions (if measure fluxes are over-estimated) or occupy SOC that does not contribute to emissions (if measurements are under-estimated). This can be analysed.

R10.3: We agree with the reviewer, that the models could not properly handle the fresh carbon. However, the reasons why the different models were unable to handle it, are not necessarily based on contributions from incorrect carbon pools, as the reviewer describes. Deni has no pools - the model has just a simple decomposition model. There was no slower pool to contribute to the emissions. Coup and DNDC had all of the fresh carbon added to the fast pool, but it was not processed in the model quickly enough. This may have been due to issues with inappropriate water management of DNDC, as described in the paper (section 4.2.2).

The decomposition approaches of the models cannot manage the ryegrass generated processes in the soil. Probably, the microbial activity increased drastically, and this had an extra speed up effect on the decomposition. The models cannot calculate the growth of the microbial community during decomposition processes (they use mostly first order kinetic). It could be one of the reasons, why the models were "too slow" and the usually applied decomposition rates of the fast pools did not work properly. This is described in the paper here:

L.516/521: "However, field and empirical data (Kuzyakov, 2010) suggest adding labile C could also enhance the decomposition of resistant pool, e.g. priming effects, which none of these models account for. Our results highlight the importance of better simulating microbial dynamics to better account for the drivers of decomposition, because these ultimately influence the denitrification flux estimations (Philippot et al., 2007). The direct application of these models with first order kinetics for decomposition to simulate the effects of fertilization or changing N deposition on denitrification fluxes could be largely biased."

Q11: Even though, I understand why the authors combined these two studies, it didn't convince me. The experiments should be separated from the modelling. This will reduce the content and make the separate analysis clearer and more specific. For both of these papers, a clear message needs to be identified and worked out by the results and discussion. The actual manuscript is not well balanced between detailed and redundant information. The is some repetition in the text that can be sorted.

**R11: While we certainly understand the comment about length, we feel strongly that presenting the information together improves the paper. The Editor of the journal agreed with us and he asked to keep the "one paper" approach, but in a shorter form. Ideally, experimental and modelling results would always be presented together. Separating the two can lead to misunderstanding (especially in the model results, if it isn't clear what modelling was based on), decreases transparency and leads to inappropriate conclusions. However, we absolutely agree that given our stance, it puts the onus on us to better edit our paper, removing the redundancies the reviewer refers to, and ensuring that the individual and overall messages are clear. We will use the suggestions from all of the reviewers to do so.**

Q12: Lines 114-115: How was the maximum water holding capacity quantified? Measured or calculated?

**R12:**

**At the silt-loam study, the soil was stored field moist (L.: 112/114)**

**At the sandy-soil the $WHC_{max}$ was determined from the measured water retention curve**

Q13: Line 190: dot missing

**R13: We add the missing dot to the sentence**

Q14: Line 202: Please keep a consistent writing of DailyDayCent.

**R14: We will change it**

Q15: Lines 496-503: This is a description of results (result section).

**R15: We reformulated the text:**

**L. 431/438: "The dramatic differences between measured fluxes of control and ryegrass soils (2-4 orders of magnitude for $CO_2$ and almost 8 for $N_2+N_2O$; Table 4) can be explained by the effects of labile carbon from ryegrass on microbial respiration and enhancement of denitrification due to increased $O_2$ consumption and supply of reductants for denitrifiers (e.g. Senbayram et al., 2018). The $CO_2$ fluxes of the ryegrass treated cores (cores 1-4)**

between days 4 and 12 show a rapid increase (Fig. S.2d). The large response of respiration to the ryegrass treatment almost hides the smaller effects resulting from the changing water and $NO_3^-$ content, while these effects were clearly visible in the control. However, small effects with a similar pattern to that seen in the control soils were also evident in the ryegrass treatments (Figs. S.1d, S.2d, S.4 day 25-35 increasing trend all cores expect core 2)."

Q16: Line 596: .and 4b

**R16: We modify the text as suggested.**

Q17: Lines 595-597: why is no pre-run or spin-up applied? This should solve the problem.

**R17: We partly agree with the reviewer, that the pre-run or spin-up could solve or cover the general problem (unknown initial parameters) of these parts of the models. The effect of the pre-run or spin-up run is similar to the calibration issue. We could have not compared the performance of the models if we had applied the pre-run or spin-up for the models. Similar to the calibration issue, we decided to use the models without pre- or spin-up run.**

**It is notable, though, that had we chosen to do a pre-run or spin-up, that this may not have solved the problem, as this is another weakness of the current models. "**

**L. 535/541: The "…coefficients originate from pure culture studies over 30 years old. These coefficients in the denitrification sub-modules (Li et al., 1992) are not universal for different soils. Here a silt-loam and sandy soil show contrasting results, suggesting that the microbial community needs soil-specific calibration. Large uncertainties in microbial coefficients must be addressed, as shown with Coup, where the denitrifier biomass was able to override the other known environmental factors for denitrification, leading to biased simulations."**

Q18: How many replicates? I think it is mentioned four, so I would expect the standard deviation (or another parameter indicating the range) in the results.

**R18: We completely agree with the critic. We add SD to Table 4.**

Q19: COUP: Which time step was used (it is mentioned that it can do hourly, but soil water content and temperature was simulated daily)?

**R19: The COUP has an option to add the daily soil water content and temperature as input data. We used this feature of the model.**

---

## Author Response (AR2)

Dear Dr. Ito,

Thank you very much for the positive response and the useful comments and suggestions.

Please, find below the list of the modifications we made:

1. The last 2 sentences from the abstract were removed.
2. Thank you for drawing our attention to the incorrect citations form and inconsistent DOI information. We added the missing comas to the citations and change the form of the DOI as required.
3. Table 3 and Table 4 were moved to the supplementary material (Table 3 -> Table S.1; Table 4 -> Table S.7)
4. Figure 3 was changed. The second axis was removed at Fig. 3d, but we would leave the second axis at Fig. 3e and f. Without the secondary axis, the pattern of the modeled fluxes would not be visible. However, we completely agree with the Editor, that the second axis is confusing. To avoid this, we extend the text and the description of Fig. 3 with the following sentences:
   a. L. 384: "However, the modeled fluxes at Coup and DNDC are definitely lower than the measured fluxes and they are not in the same magnitude (Fig. 3e and f, secondary y axis)."
   b. Fig. 3: "Important to note the difference in magnitude of the modelled and measured $N_2+N_2O$ fluxes (Fig3e and f, secondary y axis)."

Additional changes we made:

5. Line 4: remove "B." from author list
6. Line 239: "Saxton and Rawls, 2000" changed to "Saxton and Rawls, 2006".
7. Lines 77; 163; 242; 356; 368; 381; 421; 500; 783, and from the end of the description of Fig. 1 and from the end of the title of section 4.2.2 (double spaces removed).
8. Figure 1 was changed. We added the missing ticks and numbers to the x axis of d, e, f subplots.
9. We added the missing URL link to the reference of the GNUPlot program and reference a newer version (date change from 2011 to 2019 in line 260). "Williams, T. and Kelley, C.: Gnuplot 5.2: an interactive plotting program. URL: http://gnuplot.sourceforge.net/docs_5.2/Gnuplot_5.2.pdf, 2019"